# Integration of human adipocyte chromosomal interactions with adipose gene expression prioritizes obesity-related genes from GWAS

David Z. Pan[1,2], Kristina M. Garske[1], Marcus Alvarez[1], Yash V. Bhagat[1], James Boocock[1], Elina Nikkola[1], Zong Miao[1,2], Chelsea K. Raulerson[3], Rita M. Cantor[1], Mete Civelek [4], Craig A. Glastonbury[5], Kerrin S. Small [6], Michael Boehnke[7], Aldons J. Lusis[1], Janet S. Sinsheimer[1,8], Karen L. Mohlke[3], Markku Laakso[9], Päivi Pajukanta[1,2,10] & Arthur Ko [1,10]

Increased adiposity is a hallmark of obesity and overweight, which affect 2.2 billion people world-wide. Understanding the genetic and molecular mechanisms that underlie obesity-related phenotypes can help to improve treatment options and drug development. Here we perform promoter Capture Hi–C in human adipocytes to investigate interactions between gene promoters and distal elements as a transcription-regulating mechanism contributing to these phenotypes. We find that promoter-interacting elements in human adipocytes are enriched for adipose-related transcription factor motifs, such as PPARG and CEBPB, and contribute to heritability of *cis*-regulated gene expression. We further intersect these data with published genome-wide association studies for BMI and BMI-related metabolic traits to identify the genes that are under genetic *cis* regulation in human adipocytes via chromosomal interactions. This integrative genomics approach identifies four *cis*-eQTL-eGene relationships associated with BMI or obesity-related traits, including rs4776984 and *MAP2K5*, which we further confirm by EMSA, and highlights 38 additional candidate genes.

[1] Department of Human Genetics, David Geffen School of Medicine at UCLA, Los Angeles, CA 90095, USA. [2] Bioinformatics Interdepartmental Program, UCLA, Los Angeles, CA 90095, USA. [3] Department of Genetics, University of North Carolina, Chapel Hill, NC 27599, USA. [4] Department of Biomedical Engineering, University of Virginia, Charlottesville, VA 22904, USA. [5] Big Data Institute, University of Oxford, Oxford OX3 7LF, UK. [6] Department of Twin Research and Genetic Epidemiology, King's College, London, UK. [7] Department of Biostatistics, University of Michigan, Ann Arbor, MI 48109, USA. [8] Department of Biomathematics, David Geffen School of Medicine at UCLA, Los Angeles, CA 90095, USA. [9] Institute of Clinical Medicine, Internal Medicine, University of Eastern Finland and Kuopio University Hospital, FI-70210 Kuopio, Finland. [10] Molecular Biology Institute at UCLA, Los Angeles, CA 90095, USA. These authors contributed equally: David Z. Pan, Kristina M. Garske. Correspondence and requests for materials should be addressed to A.K. (email: a5ko@ucla.edu)

Obesity is a serious health epidemic world-wide. A recent study of 195 countries estimated that 2.2 billion people were overweight or obese in 2015[1]. Clinically, obesity is diagnosed by a body mass index (BMI) greater than 30. While a significant proportion of the phenotypic variation in BMI is attributed to genetic variation (heritability of BMI ~0.4–0.7[2]), understanding the mechanisms underlying this heritable component has been challenging. The 97 loci identified in a genome-wide association study (GWAS) for BMI in ~340,000 subjects explain only 2.7% of the variance in BMI, and all HapMap phase 3 genetic variants (~1.5 M single nucleotide polymorphisms (SNPs)) were estimated to account for ~21% of the variance in BMI in 16,275 unrelated individuals[2]. The causal variants and genes are not immediately apparent from GWAS, hindering our ability to understand the biological mechanisms by which genetics contribute to obesity. To address this knowledge gap, we integrate chromosomal interaction data from primary human white adipocytes (HWA) with adipose gene expression and clinical phenotype data (BMI, waist-hip ratio, fasting insulin, and Matsuda index) to elucidate molecular pathways involved in genetic regulation in cis.

Combining genotype and RNA-sequencing (RNA-seq) data enables the detection of expression quantitative trait loci (eQTLs) that regulate transcription of near-by genes (i.e., in cis). These cis-eQTLs often reside in regulatory elements, including promoters, enhancers, and super-enhancers. However, the mechanism by which cis-eQTLs regulate their respective eGene(s) is seldom established because identification of the true regulatory variants among SNPs in tight linkage disequilibrium (LD) has proven challenging[3]. Enhancers modulate target gene expression levels via their interaction with promoters, and disruption or improper looping of enhancer sites can contribute to disease risk[4,5]. Promoter Capture Hi–C (pCHi–C) enables detection of promoter interactions at a higher resolution and at lower sequencing depth than that required for Hi–C[6]. Incorporating a chromosomal interaction map constructed from pCHi-C and cis-eQTL data can help elucidate the functional mechanisms by which the genetic variants affect gene expression. By overlapping these looping cis-eQTLs with trait-associated variants identified in independent, large-scale GWAS, we can assess which GWAS variants could affect expression of regional genes via chromosomal interactions.

To search for genes that are functionally important for adipose tissue biology, we performed a cis-eQTL analysis using genome-wide SNP data and adipose RNA-seq data from individuals of the Finnish METabolic Syndrome In Men (METSIM) cohort. We identified 42 genes, regulated by cis-eQTLs that reside in regions that physically interact with the promoters of genes. Adipose expression of these 42 genes was robustly correlated with BMI, and among them four genes, MAP2K5, LACTB, ORMDL3, and ACADS, were regulated by SNPs (or their tight LD proxies) previously identified in GWAS for BMI or a related metabolic trait, located at the regulatory element-promoter interaction sites. These data provide converging evidence for effects of looping cis-eQTL variants on gene expression associated with obesity and related metabolic traits. Our results show that these integrative genomics methods involving pCHi-C data in primary HWA can identify regulatory circuits comprising both regulatory elements and their target gene(s) that operate in a complex obesity-related metabolic trait.

## Results

### Characterization of the adipocyte chromosomal interactions.
Adipose tissue is highly heterogeneous, containing adipocytes, preadipocytes, stem cells, and various immune cells. We performed pCHi-C in primary HWA with the goal of identifying physical interactions between adipose cis-eQTLs and target gene promoters. We employed the pCHi-C protocol as described previously[7]. Briefly, we fixed primary HWA to crosslink proteins to DNA, and after digestion with the HindIII restriction endonuclease, we performed in-nucleus ligation and biotinylated RNA bait hybridization to pull down only those HindIII fragments with annotated gene promoters[6]. To detect the regions that interact with the promoter-containing HindIII fragments, we mapped the reads to the genome, and assigned reads to HindIII fragments to allow for fragment-level resolution of those regions interacting with the baited fragments containing gene promoters. The key pCHi-C sequencing metrics are shown in Supplementary Table 1.

We first confirmed that the non-promoter regions in adipocyte chromosomal interactions are enriched for enhancer (H3K4me1, H3K4me3, and H3K27ac), repressor (H3K27me3, H3K9me3) histone marks, and DNase I hypersensitive sites (DHSs) (Supplementary Table 2). As there are no publicly available DHS data for adipocytes or adipose tissue, we used the union of DHSs in all cell types from ENCODE and Roadmap rather than DHSs in a single, non-adipocyte cell type[8]. Intersecting the adipocyte and previously published primary CD34+ cell pCHi-C data[6], we found that 68.0% of adipocyte pCHi-C chromosomal interactions were observed in adipocytes but not in CD34+ cells. In the following, we used the same public DHS data to focus on open chromatin regions as they are more likely to bind transcription factors (TFs) and, thus, be relevant for chromosomal looping interactions within the interacting HindIII fragments.

We examined whether the DHSs are enriched for adipose-related TF motifs, using the Hypergeometric Optimization of Motif EnRichment (HOMER) software[9] that calculates the number of times a TF motif is seen in target and background sequences. The proportion of times the TF motif is seen in the target when compared to the background is then tested for enrichment in the target sequences. We found that when compared to DHSs within CD34+ chromatin interactions, the DHSs within the adipocyte chromatin interactions are enriched for 26 of 332 TF motifs (FDR < 5%) (Supplementary Table 3), including CCAAT/enhancer binding protein beta (CEBPB, $p$-value $= 1.00 \times 10^{-10}$) and peroxisome proliferator-activated receptor gamma (PPARG, $p$-value $= 0.01$), both of which are well-known key players in adipose biology[10]. To address the potential bias of using a different pCHi-C dataset as background, we also performed HOMER analysis comparing the DHSs in adipocyte interactions to DHSs in non-interacting, non-promoter regions in the remainder of the genome. The results were similar, and both CEBPB and PPARG were also enriched in the latter analysis (CEBPB, $p$-value $= 1.00 \times 10^{-24}$; PPARG, $p$-value $= 1.00 \times 10^{-6}$; complete enrichment results not shown). These results suggest that the cell-type based pCHi-C interaction data enable the detection of interactions important for that cell type within a heterogeneous human tissue.

**Chromosomal interactions explain expression heritability**. To investigate whether the variants residing within open chromatin of chromosomal looping regions in adipocytes are enriched for SNPs that contribute to the heritability of cis expression regulation, we partitioned the heritability of cis regulation of human adipose gene expression into 52 functional categories using a modified partitioned LD Score regression method[11] (see Methods). The 52 functional categories are derived from 26 main annotations that include coding regions, untranslated regions (UTRs), promoters, intronic regions, histone marks, DNase I hypersensitivity sites (DHSs), predicted enhancers, conserved regions, and other annotations[11] (Supplementary

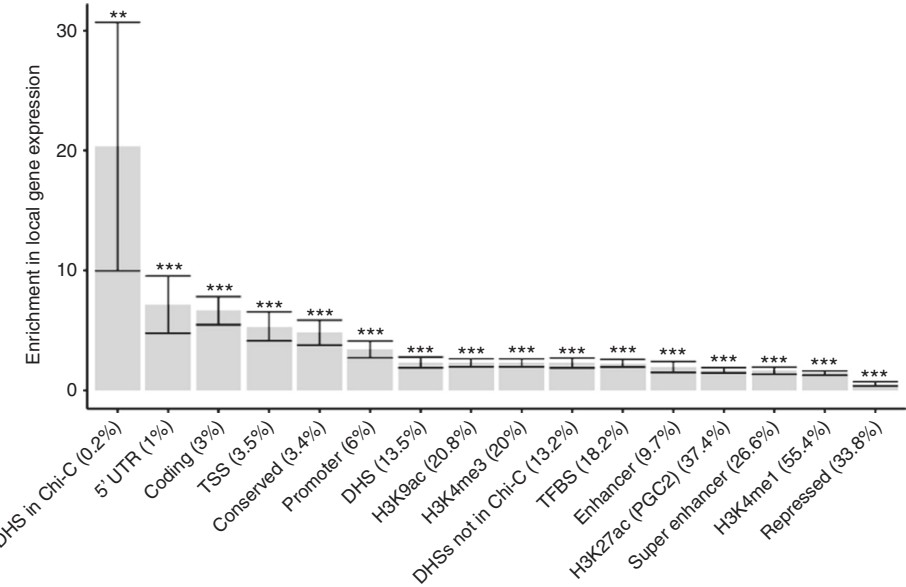

**Fig. 1** Open chromatin sites (DHSs) within adipocyte promoter CHi-C chromosomal interactions show significant enrichment in *cis* expression. Enrichments in *cis* expression with error bars for different categories using LD score regression analysis (see Methods). For the horizontal axis labels, the value in parentheses shows the percentage of SNPs contained within the respective annotation category that contributed to the enrichment calculation. For the significance threshold after Bonferroni correction above each bar, * indicates a *p*-value < 0.05; **, a *p*-value < 0.001; and ***, a *p*-value < 0.0001, respectively. The *p*-values were estimated based on Z scores calculated from the normal distribution. Error bars represent jackknife standard errors around the estimates of enrichment

Figure 1, Supplementary Tables 4–5). The partitioned LD Score regression method[11] utilizes summary association statistics of all variants on gene expression to estimate the degree to which variants in different annotation categories explain the heritability of *cis* and *trans* expression regulation while accounting for the LD among functional annotations. To assess the enrichment of heritability mediated by the variants in the chromosomal interactions detected by pCHi-C on a per-gene basis, we further modified the LD score method, as described in detail in the Methods. Importantly, these modifications did not change the 52 baseline enrichments significantly when compared with the data obtained using the unmodified version[11] (Supplementary Figure 1, Supplementary Tables 4–5). These analyses revealed that open chromatin regions (i.e., DHSs) within the adipocyte chromosomal interactions are enriched for sequences that contribute to heritability of gene expression regulation in *cis* (Fig. 1, *p*-value < 0.002, enrichment = 20.3 (SD±5.2), average proportion of SNPs = 0.23%). The variants residing within the open chromatin regions within adipocyte chromosomal interactions explain 4.6% of the heritability of adipose tissue gene expression in *cis*, despite only accounting for 0.23% of the SNPs per *cis* gene region on average, indicating the functionality of these SNPs at the DHSs of distal interactions in regulating *cis* expression.

**Identification of genes regulated by looping *cis*-eQTL SNPs.** To identify adipose-expressed genes regulated by SNPs (eGenes), we performed a *cis*-eQTL analysis using 335 individuals from the METSIM cohort with both genome-wide SNP data and adipose RNA-seq data available (Fig. 2; Methods). Using the published adipose *cis*-eQTL data and criteria for significance from GTEx[12] (see Methods), we found 487,679 *cis*-eQTLs for 4,650 eGenes in the METSIM dataset and confirmed these same SNPs as *cis*-eQTLs by look-up in GTEx. 386,068 of the 487,679 (79.0%) *cis*-eQTL SNPs had the same target gene and direction of effect in both cohorts (Supplementary Figure 2). Only the 386,068

*cis*-eQTL SNPs that were replicated for effect direction and target gene (Supplementary Table 1) in the GTEx adipose RNA-seq data were used in our subsequent downstream analyses (Supplementary Figure 2). Overall, 4,332 of 4,650 of *cis*-eQTL-eGene relationships (93.0%) were replicated using the published adipose *cis*-eQTL data and criteria for significance from GTEx[12] (see Methods). To restrict these adipose *cis*-eQTL SNPs to those that likely function through transcription factor (TF) binding at distal regulatory elements, we determined which of these eGene promoters were involved in looping interactions with the *cis*-eQTLs, assayed through pCHi-C in primary HWA (Fig. 2; Methods). Of the 4,332 eGenes identified in our *cis*-eQTL analysis, 576 (13.4%), were involved in these looping interactions (permutation *p*-value < 0.00001) (Fig. 2, Methods, Supplementary Figure 2, and Supplementary Table 1).

We next determined the set of 576 looping eGenes with expression levels that are correlated with BMI in METSIM (Pearson correlation, adjusted $p < 1.15 \times 10^{-5}$ to correct for the 4,332 replicated eGenes identified in our *cis*-eQTL analysis). We found 54 of 576 (9.40%) BMI-correlated eGenes with promoters involved in looping interactions with their *cis*-eQTL SNP (Supplementary Table 6). In our subsequent second replication analysis, the expression levels of 42 out of 54 genes (replication rate of 77.8%) were correlated with BMI in adipose RNA-seq data from the TwinsUK cohort (*n* = 720) with the same direction of effect on BMI as in METSIM (Bonferroni adjusted *p* < 0.001) (Table 1, Supplementary Table 6). Another four of the 54 genes were not available in the TwinsUK dataset. The effects sizes and *p*-values obtained for BMI associations in TwinsUK and METSIM, using a linear regression model in both, show comparable results to those obtained using the Pearson correlations (Table 1, Supplementary Table 6). These 42 BMI-correlated genes are functionally enriched for four pathways with fatty acid metabolism as a top ranking pathway (Supplementary Table 7) based on KEGG pathway enrichment using WebGestalt[13] (Benjamini-Hochberg adjusted *p* < 0.05); however, the small

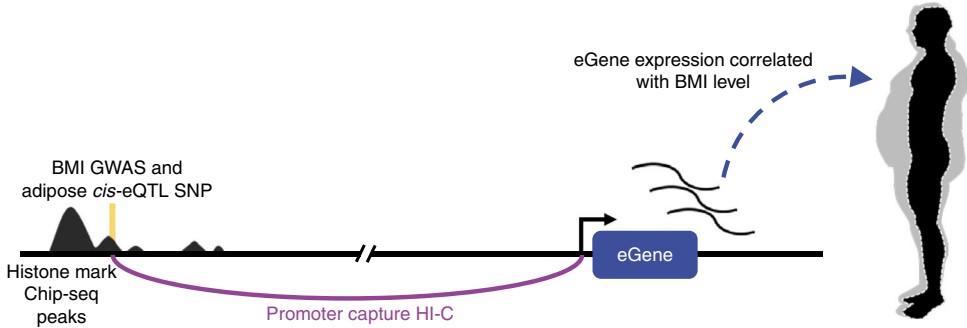

**Fig. 2** Overview of the study design targeted to identify new genes for obesity and related metabolic traits. A schematic illustrating the integration of multi-omics data utilized in this study to elucidate genetics of obesity-related traits.

number of genes in these pathway analyses warrant verification in future studies. Only these 42 replicated genes were further investigated in our downstream analyses.

**Adipocyte chromosomal interactions prioritize GWAS genes.** To investigate which of the 42 BMI-correlated eGenes are regulated by GWAS variants previously identified for BMI and related metabolic traits, we determined which interacting *cis*-eQTL variants are GWAS variants (or their LD proxies, $r^2 > 0.80$), using $p < 5.00 \times 10^{-8}$ as a criterion to select variants. As the goal of the current study was to dissect the molecular contribution of adipose and adipocyte biology to traits that can influence the pathophysiology of obesity, we examined GWAS for BMI and the metabolic traits that have previously been shown to exhibit co-morbidities with obesity (e.g., serum lipids and type 2 diabetes) or that are influenced by obesity or correlated with BMI (e.g., metabolites and WHR). We used all GWAS variants ($p$-value < $5.00 \times 10^{-8}$) identified in a previous metabolite GWAS of ~7000 individuals[14], lipid GWAS of ~180,000 individuals[15], an extensive BMI GWAS study of ~340,000 individuals[2], a sequencing-based GWAS for type 2 diabetes[16], and a waist-hip-ratio (WHR) adjusted for BMI GWAS of ~220,000 individuals[17]. We found a GWAS variant for BMI, regulating mitogen-activated protein kinase kinase 5 (*MAP2K5*); a GWAS variant for high-density lipoprotein cholesterol (HDL-C), regulating orosomucoid like sphingolipid biosynthesis regulator 3 (*ORMDL3*); and two GWAS variants for serum metabolites (succinylcarnitine and butyrylcarnitine), regulating lactamase beta (*LACTB*) and acyl-CoA dehydrogenase, C-2 To C-3 short chain (*ACADS*), among the 42 genes (Fig. 3a, b; Supplementary Figure 3a–f), with the looping interactions spanning 287 kb, 16 kb, 151 kb, and 183 kb, respectively. We found that the interacting *cis*-eQTL-containing *Hind*III fragments for *LACTB* and *MAP2K5* are located within the promoter and intron of other genes. Furthermore, using the integrated pCHi-C and *cis*-eQTL data, we found that the SNPs in these regulatory *Hind*III fragments regulate genes that are not their nearest gene for 3 of the 4 BMI-correlated eGenes (Fig. 3a, b, Supplementary Figure 3a–f).

**The looping BMI GWAS SNPs regulate *MAP2K5*.** For *MAP2K5*, the reported BMI GWAS SNP itself is not located within the regulatory, *cis*-eQTL-containing *Hind*III fragment involved in the looping interaction; however, SNPs in tight LD with the GWAS SNP (using a criterion of $r^2 > 0.80$) are in the regulatory *Hind*III fragment that is interacting with the target gene promoter (Fig. 3b). The regulatory *Hind*III fragment contains 16 *cis*-eQTL SNPs that are LD proxies for the BMI GWAS SNP[2] (rs16951275), which has a total of 62 LD proxies in the

METSIM cohort. To prioritize a candidate functional variant within these 16 SNPs within the *Hind*III fragment, we first examined the predicted TF motifs that may be affected by each SNP using the data curated from ChIP-seq by Kheradpour and Kellis[18]. We found that only rs4776984, which is in almost perfect LD with the original BMI GWAS variant rs16951275 ($r^2 = 0.98$), showed a predicted increase in binding of CTCF, which is a TF known to mediate chromosomal interactions (Fig. 4a).

We also used the deep learning–based sequence analyzer (DeepSEA)[19] to examine the allelic effect on protein binding of rs4776984 and the 15 other looping *cis*-eQTLs of *MAP2K5*. Of these 16 looping *cis*-eQTLs, six were potentially functional and of these, two variants passed the functional significance score of <0.05 using DeepSEA. Of the two, our candidate functional eQTL SNP, rs4776984, resulted in the most significant functional score ($2.36 \times 10^{-3}$) (Supplementary Table 8) and was the only variant passing a functional significance score of <0.01 among the 16 variants. Thus, the DeepSEA result further supports the differential TF binding at the variant site rs4776984 among all possible looping *cis*-eQTLs at the *MAP2K5* locus (Supplementary Table 8). The looping *cis*-eQTL site also shows a ChIP-seq peak for the histone mark H3K4me1 in ENCODE adipose nuclei ChIP-seq data; however, notably it also shows the presence of the histone marks H3K27me3 and H3K9me3 (Fig. 3b), two marks known to be associated with transcriptional repression. Furthermore, the gene expression of *MAP2K5* is negatively correlated with BMI ($p$-value = $7.83 \times 10^{-6}$). These data implicate *MAP2K5* as a gene regulated by the BMI GWAS signal via a repressive chromosomal interaction.

To functionally assess whether there is differential allele-specific binding of proteins at the candidate functional *MAP2K5* eQTL, rs4776984, we performed electrophoretic mobility shift assays (EMSAs) using nuclear protein from primary HWA. The results show reduced protein binding of the reference allele when compared to the alternate allele of rs4776984, consistently in three independent experiments (Fig. 4b, Supplementary Figure 4), in line with the predicted disruption in protein binding for CTCF[18] (Fig. 4a). We performed the supershift experiment using the CTCF antibody and adipocyte nuclear extract, but did not observe a supershift in any of the three replicated experiments (Supplementary Figure 5). We repeated the supershift experiment using a different CTCF antibody (EMD Millipore 07–729), which resulted in the same negative finding (Supplementary Figure 6). To further verify the negative supershift result, we also directly tested the CTCF protein for allele-specific binding at rs4776984 using EMSA in 3 replicate experiments, and did not find evidence of sole CTCF protein binding (Supplementary

**Table 1 Thirteen representative eGenes (9 most significant genes and 4 GWAS loci) that correlate with BMI in METSIM and TwinsUK (for the full data on all 54 genes, see Supplementary Table 6)**

| Rank[a] | Gene | Chr[f] | Pearson | | Linear regression | | | | | |
| | | | METSIM[c] | | METSIM[d] | | | TwinsUK[e] | | |
| | | | Effect size (r) | p-value | Effect size (β) | SE | p-value | Effect size (β) | SE | p-value |
|---|---|---|---|---|---|---|---|---|---|---|
| 1 | *ADH1B* | 4 | −0.45 | $7.40 \times 10^{-18}$ | −0.21 | 0.02 | $1.68 \times 10^{-20}$ | −0.58 | 0.03 | $4.47 \times 10^{-71}$ |
| 2 | *ORMDL3*[b] | 17 | −0.45 | $8.57 \times 10^{-18}$ | −0.16 | 0.02 | $2.06 \times 10^{-20}$ | −0.58 | 0.03 | $2.65 \times 10^{-70}$ |
| 3 | *AKR1C3* | 10 | 0.33 | $4.78 \times 10^{-10}$ | 0.13 | 0.02 | $2.95 \times 10^{-11}$ | 0.49 | 0.03 | $5.19 \times 10^{-54}$ |
| 4 | *CMTM3* | 16 | 0.41 | $4.32 \times 10^{-15}$ | 0.087 | 0.01 | $3.84 \times 10^{-17}$ | 0.50 | 0.03 | $6.64 \times 10^{-52}$ |
| 5 | *LPIN1* | 2 | −0.38 | $1.49 \times 10^{-13}$ | −0.14 | 0.02 | $2.27 \times 10^{-15}$ | −0.47 | 0.03 | $2.38 \times 10^{-44}$ |
| 6 | *RNF157* | 17 | −0.29 | $5.19 \times 10^{-8}$ | −0.096 | 0.02 | $5.87 \times 10^{-9}$ | −0.47 | 0.03 | $8.86 \times 10^{-42}$ |
| 7 | *MYOF* | 10 | 0.32 | $1.07 \times 10^{-9}$ | 0.086 | 0.01 | $7.37 \times 10^{-11}$ | 0.46 | 0.03 | $2.59 \times 10^{-40}$ |
| 8 | *NAA40* | 11 | 0.28 | $1.81 \times 10^{-7}$ | 0.052 | 0.009 | $2.67 \times 10^{-8}$ | 0.46 | 0.03 | $4.00 \times 10^{-40}$ |
| 9 | *TMEM165* | 4 | 0.33 | $2.45 \times 10^{-9}$ | 0.045 | 0.007 | $1.84 \times 10^{-10}$ | 0.45 | 0.03 | $3.52 \times 10^{-37}$ |
| 10 | *RFFL* | 11 | 0.27 | $1.02 \times 10^{-6}$ | 0.035 | 0.006 | $1.84 \times 10^{-7}$ | 0.43 | 0.03 | $5.67 \times 10^{-37}$ |
| 28 | *ACADS*[b] | 12 | −0.37 | $2.91 \times 10^{-12}$ | −0.085 | 0.01 | $7.12 \times 10^{-14}$ | −0.24 | 0.03 | $6.65 \times 10^{-19}$ |
| 31 | *LACTB*[b] | 15 | 0.30 | $1.67 \times 10^{-8}$ | 0.069 | 0.01 | $1.40 \times 10^{-9}$ | 0.32 | 0.04 | $4.94 \times 10^{-18}$ |
| 34 | *MAP2K5*[b] | 15 | −0.25 | $7.83 \times 10^{-6}$ | −0.039 | 0.01 | $1.90 \times 10^{-6}$ | −0.21 | 0.03 | $3.81 \times 10^{-10}$ |

[a] Thirteen representative eGenes, including 4 GWAS loci, ranked by their p-value in the TwinsUK cohort dataset
[b] GWAS gene
[c] Effect size (r, Pearson rho) and p-value calculated from Pearson correlation between gene expression and BMI (see Methods)
[d] Effect size, standard error (SE), and p-value calculated using a linear regression model with BMI and age, age² and the 14 technical factors as covariates when compared to a null model without BMI. These models were compared using an F-test (see Methods)
[e] Effect size, standard error (SE), and p-value calculated from linear mixed effects model. A full model including BMI was compared to a null model in which the same model was fitted, but with BMI omitted. These models were compared using an F-test (see Methods)
[f] Chr is an abbreviation for chromosome

Figure 7). However, a supershift experiment may remain negative even in the presence of true TF binding if a complex instead of a single TF alone is required for the TF binding[20].

**Interacting GWAS SNPs implicate three other genes**. For *ORMDL3*, there is a single lipid GWAS SNP, rs8076131, in the *Hind*III fragment, which is also the only looping *cis*-eQTL SNP interacting with the *ORMDL3* promoter. Variant rs8076131 lies in a region with enhancer histone marks H3K4me1 and H3K27ac in adipose nuclei (Supplementary Figure 3a,b). The expression of *ORMDL3* is negatively correlated with BMI ($p = 8.57 \times 10^{-18}$), in line with its known role as a negative regulator of sphingolipids that are positively correlated with obesity[21,22].

The regulatory *Hind*III fragment that loops with the *LACTB* promoter contains two reported metabolite GWAS SNPs in tight LD with each other (rs1017546 and rs3784671, $r^2 = 0.97$), both sharing 35 LD proxies ($r^2 > 0.80$) in the METSIM cohort. One of the two index GWAS SNPs within the *Hind*III fragment, rs3784671, resides in a region enriched for the enhancer histone marks H3K4me1 and H3K27ac in adipose nuclei (Supplementary Figure 3c, d). This metabolite GWAS SNP, rs3784671, is associated with succinylcarnitine levels, which have previously been shown to be positively correlated with BMI in KORA ($p = 1.0 \times 10^{-12}$) and TwinsUK ($p = 5.3 \times 10^{-5}$)[23]. Accordingly, the expression of *LACTB* is positively correlated with BMI ($p = 1.19 \times 10^{-8}$). Notably, *LACTB* has been implicated as a causal gene for obesity in mice[24], further supporting our integrated human data that implicates LACTB involvement in an obesity-related metabolic trait.

The most significant metabolite GWAS SNP for *ACADS*, rs10774569, is not located within the regulatory, *cis*-eQTL-containing *Hind*III fragment. Instead, a single *cis*-eQTL SNP rs12310161, in perfect LD ($r^2 = 1.0$) with the GWAS SNP rs10774569, is the only *cis*-eQTL SNP located within the regulatory *Hind*III fragment, looping with the fragment containing the promoter of *ACADS*. This looping *cis*-eQTL SNP also resides in a region enriched for enhancer histone marks H3K4me1 and H3K27ac in adipose nuclei (Supplementary Figure 3e, f). The expression of *ACADS* has a negative correlation with BMI ($p = 2.91 \times 10^{-12}$), and the alternate allele is associated with an increase in expression of *ACADS*, suggesting that this allele has a protective effect against obesity.

Finally, we repeated the pCHi-C experiments in the same HWA cell line in a separate experiment with two replicates and found the same GWAS SNP interactions as in the first experiment (Supplementary Table 9). This validation data thus provides further support for our conclusions and the robustness of interactions we report.

## Discussion

BMI is a highly complex trait caused by the poorly characterized interplay between genetic and environmental factors with upper heritability estimates reaching 70%[2]. Understanding how genome-wide signals with small effect sizes contribute to BMI on a molecular level has proven to be difficult. Delineating the underlying biological mechanisms of these signals is crucial to better understand the development of obesity and its concomitant cardiometabolic disorders. In this study, we performed promoter Capture Hi–C (pCHi–C) in primary human white adipocytes (HWA) to identify BMI-correlated adipose-expressed genes that are under genetic regulation in *cis* by variants that physically interact with gene promoters. Through our method of integrating GWAS, *cis*-eQTL analyses, chromosomal interactions, and robust replication of the data from GTEx and TwinsUK, we were able to identify 42 candidate genes for future obesity research.

In the absence of adipocyte DHS information, we used DHS data from all tissues in the ENCODE and Roadmap Epigenomics project to label open chromatin regions within the adipocyte chromosomal interactions[8]. Despite this methodological compromise, our results demonstrate that variants in these regions

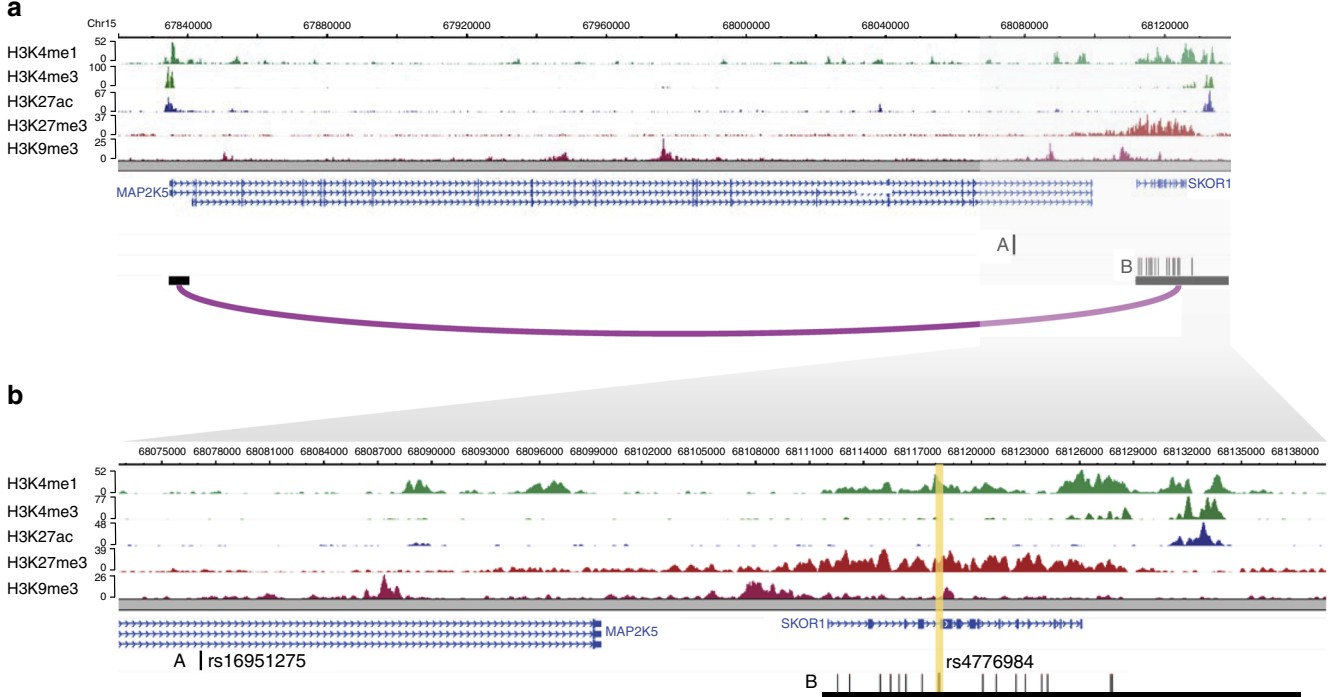

**Fig. 3** Promoter Capture Hi–C enables refinement of the BMI GWAS locus that colocalizes with *cis*-eQTLs interacting with the target gene promoter of *MAP2K5*. Genomic landscape of the BMI locus, *MAP2K5* (panels **a**, **b**), modified from the WashU Genome Browser to show the histone mark calls from ChIP-seq data; gene transcripts; promoter and eQTL *Hind*III fragments that interact in primary human white adipocytes (HWA); and GWAS SNPs (A, the rs number indicated in the magnified box) or their LD proxies (B, $r^2 > 0.8$) located in the interacting *Hind*III fragment. The vertical yellow band highlights the *cis*-eQTL variant (the rs number is indicated in the magnified box). **a** Genomic landscape containing *MAP2K5* and the interacting *cis*-eQTL variant and corresponding BMI GWAS SNP. **b** Magnification of the boxed region in (**a**)

explain a significant portion (4.6%) of the heritability of *cis*–regulated expression in human subcutaneous adipose tissue. Even though the total percentage of variants within the intersection of open chromatin regions and adipocyte chromosomal looping sites is small (0.23%), the enrichment implies that these SNPs are functionally relevant for adipocyte biology and gene regulation in *cis*.

The enrichment of TF binding motifs for CEBPB and PPARG in chromosomal interactions found in adipocyte but not in CD34[+] cells confirms that the regulatory circuits identified here are relevant to adipose biology. These two TFs have previously been shown to occupy shared regulatory sites. Apart from being an enhancer binding protein, which is in concordance with its presence at chromosomal interaction sites, CEBPB has been demonstrated to precede the binding of PPARG at many regulatory sites[25], suggesting that CEBPB primes the regulatory regions for the binding of the adipose master regulator PPARG.

One of our looping *cis*-eQTL variants is a tight LD proxy ($r^2 = 0.98$) for a regional BMI lead GWAS SNP (rs16951275)[2]. Typical fine mapping techniques such as overlaying histone marks, transcription factor motif scans, or eQTL searches do not necessarily reveal the mechanism through which a SNP might function. We refined the GWAS signal from 64 to 16 LD SNPs within a *Hind*III fragment that interacts with the *MAP2K5* promoter by overlaying *cis*-eQTLs, the promoter-enhancer interaction map, and the expression-BMI correlation. The top candidate, rs4776984, increased HWA nuclear protein binding in an allele-specific way in our EMSA experiment and lies within the repressor histone marks H3K27me3 and H3K9me3 in ENCODE adipose nuclei data. Recent studies have suggested that repressor elements function through looping interactions in a similar manner to enhancer elements[6,26], which would align well with the

negative correlation between expression of *MAP2K5* and BMI level.

The region at the *MAP2K5* locus, exhibiting increased binding for the alternate allele for rs4776984, contains predicted motifs for the looping interaction protein, CTCF, and other TFs (Supplementary Table 8). We did not find evidence of CTCF binding at rs4776984 in our supershift and protein binding EMSA experiments. However, a supershift experiment may remain negative even in the presence of true TF binding if a complex instead of a single TF alone is required for the TF binding[20]. Furthermore, using DeepSEA analysis, we confirmed the potential for differential TF binding at the variant site rs4776984 among all possible looping *cis*-eQTLs at the *MAP2K5* locus. Noteworthy, since DeepSEA identified multiple TFs as potential binders of rs4776984 site in an allele-specific way, future studies testing a larger set of TFs are warranted to identify the actual TF that binds this site. We postulate that TF binding at this interaction site would lead to a repressive looping mechanism, in this case altering *MAP2K5* expression in adipocytes.

MAP2K5 is a member of the ERK5 MAP kinase signaling cascade, and the importance of ERK5 signaling in adipose was previously demonstrated in *Erk5* knock-out mice, which exhibit increased adiposity[27]. This suggests that changes in ERK5 signaling in adipocytes could be relevant for human obesity. MAP2K5 is a strong and specific activator of ERK5 in the ERK5 MAP kinase signaling cascade[28], supporting further study of *MAP2K5* in connection with increased adiposity.

The intronic *ORMDL3* GWAS variant rs8076131 is associated with high-density lipoprotein cholesterol (HDL-C)[15] and is the only *cis*-eQTL SNP in the *Hind*III fragment that interacts with the *ORMDL3* promoter in our adipocyte pCHi-C data. ORMDL3 is a negative regulator of the synthesis of sphingolipids that are

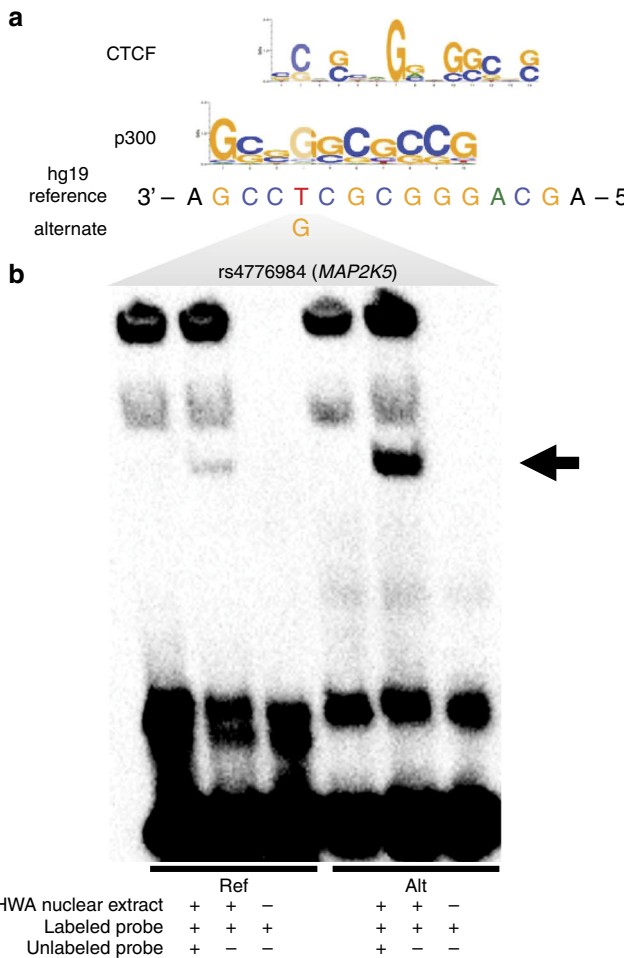

**Fig. 4** Predicted TF motifs and electrophoretic mobility shift assay (EMSA) at the rs4776984 site indicate allele-specific binding. **a** Predicted TF motifs for CTCF and p300, as well as the hg19 reference genome sequence. **b** Biotinylated (labeled probe) 31-bp oligonucleotide complexes with ±15 bp flanking the reference or alternate allele for variant rs4776984 were incubated with nuclear protein extracted from primary HWA and resolved on a 6% polyacrylamide gel. Competitor assays were performed by incubating the reaction with ×100 excess of unlabeled (no biotin) oligonucleotide complexes with identical sequence. Arrow denotes specific binding of HWA nuclear protein to reference (left) and alternate (right) allele

produced in response to obesity and related metabolic traits, such as inflammation and insulin resistance[21,22], and that interfere with important signaling pathways associated with these traits[22]. Corroborating this, we show that *ORMDL3* expression is negatively correlated with BMI, and the *cis*-eQTL and risk variant rs8076131 decreases *ORMDL3* expression, potentially through a change in the chromosomal interaction between the enhancer and promoter of *ORMDL3*, as has been shown for this enhancer site previously[29].

We found that the metabolite GWAS SNP, rs3784671, is a looping *cis*-eQTL variant associated with the expression levels of the *LACTB* gene. Although this variant is a *cis*-eQTL for *LACTB* both in our study and the GTEx adipose cohort, it lies within the promoter for the *APH1B* gene, for which it is not a *cis*-eQTL in our study. Through overlap of adipose *cis*-eQTL data and adipocyte pCHi-C data, we established that rs3784671 does not act through the adjacent *APH1B* gene and filtered the 35 *cis*-eQTL variants for *LACTB* down to a single variant, rs3784671. This

variant is negatively associated with the levels of succinylcarnitine, a metabolite positively correlated with BMI in two independent cohorts, KORA and TwinsUK, previously[23]. Succinylcarnitine is a molecule in the butanoate metabolism pathway; butanoate has been implicated in anti-inflammation, protection against obesity, and an increase in leptin levels[30]. Furthermore, as the succinylcarnitine GWAS variant rs3784671 is an eQTL for *LACTB*, associated with an increase in *LACTB* expression, we postulate that *LACTB* expression increases succinylcarnitine. This is in agreement with a mouse study that shows that butanoate metabolism is reduced in *Lactb* transgenic mice[24]. Notably, support for *LACTB* as a causal gene for obesity derives from functional studies using transgenic overexpression of *Lactb* in mice, resulting in an increase in the fat-mass-to-lean-mass ratio[24,31]. Although the function of LACTB in adipose has not been fully elucidated, these studies suggest that a reduction in *LACTB* function and, in turn, an increase in butanoate metabolism and decrease of succinylcarnitine levels are beneficial for obesity treatment. Further molecular studies at the protein level are, however, required to determine the function of *ORMDL3* and *LACTB* in connection with obesity.

We identified a perfect LD proxy for a metabolite GWAS SNP that lies within a *Hind*III fragment that regulates the *ACADS* gene and interacts with its promoter. ACADS is a mitochondrial protein that catalyzes the first step of the fatty acid beta-oxidation pathway. Proper mitochondrial function is imperative for adipose function and energy homeostasis. In addition to the METSIM and TwinsUK adipose RNA-seq data sets used in our study, a previous study identified *ACADS* when systematically searching for genes over and under-expressed in obese versus lean adipose tissue[32]. Furthermore, all 3 datasets show a consistent negative correlation between *ACADS* expression and BMI, in support of its well-established mitochondrial function. The interacting *cis*-eQTL and GWAS SNP, rs12310161, is located within enhancer histone marks in adipose nuclei and in the HepG2 liver cell line, with the alternate allele exhibiting a positive effect on gene expression, in line with it being a protective allele. Interestingly, this variant falls within a TEA Domain Transcription Factor 4 (TEAD4) ChIP-seq peak in the HepG2 cells. *TEAD4* expression is regulated by Peroxisome Proliferator Activated Receptor alpha (PPARα)[33], the major regulator of beta-oxidation of fatty acid pathways in liver and brown adipose tissue. Taken together, these results suggest that the interacting *cis*-eQTL and metabolite GWAS SNP, rs12310161, functions within an enhancer to increase *ACADS* expression and mitochondrial fatty acid beta-oxidation in adipose.

As the pCHi-C experiments were performed in primary HWA, we are able to focus on physical chromosomal interactions directly in human adipocytes among all cell types present in adipose tissue. Adipocytes perform central adipose functions, including lipogenesis and lipolysis. Further investigation of the adipose genes, which are under *cis* genetic regulation via chromosomal looping to the promoters and are correlated with BMI, is likely to provide much needed insight into cellular processes contributing to obesity. Our data provide 38 new candidate genes, including some known functionally relevant genes for adiposity such as *LPIN1*[34] and *AKR1C3*[35], that have so far not been highlighted by GWAS for BMI or obesity-related metabolic traits. We postulate that identification of some of these 38 candidates as obesity GWAS genes may require much larger GWA studies, while others may represent genes responding to obesity in human adipose tissue. Our analysis of the looping *cis*-eQTLs for other GWAS traits correlated with BMI, such as serum metabolites and lipids, led to the identification three additional obesity-related metabolic GWAS genes. We recognize that brain and other

tissues likely account for some of the BMI GWAS signals and that GWAS variants may act via other mechanisms, such as *trans* regulation and alternative splicing, that warrant future investigation. Although the four looping *cis*-eQTL variants identified at GWAS loci in our study represent either the GWAS tag SNPs (as is the case at the *ORMDL3* and *LACTB* loci) or they are in perfect or almost perfect LD with the GWAS SNP ($r^2 = 1.0$ at the *ACADS* locus and $r^2 = 0.98$ at the *MAP2K5* locus), we recognize that the looping variants may not always be the strongest *cis*-eQTL SNPs at these loci and, thus, additional fine mapping is needed to fully elucidate all functional regulatory *cis*-eQTL variants.

The current study uses the integration of multi-level genomic and functional data to enhance the understanding of genome-wide molecular signals underlying obesity. GWAS signals often fall within non-coding regulatory regions of the genome, and the affected gene(s) often remain unclear. Similarly, the local LD structure frequently hinders the identification and functional characterization of the actual eQTL SNP even though the eQTL target gene is known. Through the integration of multi-layer genomics data in a functionally relevant human cell type and tissue and replication in the GTEx and TwinsUK cohorts, we show that the DHSs within the interacting chromosomal regions are enriched for tissue-specific TF motifs and explain a significant proportion of the heritability of gene expression in *cis*. Furthermore, we identified *LACTB*, *ACADS*, *ORMDL3*, and *MAP2K5* as obesity-related genes in humans and provide a set of 38 non-GWAS candidate genes for future studies in obesity.

## Methods

**Cell lines and culture reagents**. We obtained and cultured the primary human white preadipocyte (HWP) cells as recommended by PromoCell (PromoCell C-12731, lot 395Z024) for preadipocyte growth and differentiation into adipocytes. Cell media (PromoCell) was supplemented with 1% penicillin-streptomycin. We maintained the cells at 37 °C in a humidified atmosphere at 5% $CO_2$. We serum-starved the primary human white adipocytes (HWA) for 16 h using 0.5% fetal calf serum (FCS) in supplemented adipocyte basal medium (PromoCell), prior to treatment with 0.23% fatty acid free bovine serum albumin (BSA, Sigma Aldrich A8806) in media containing 0.5% FCS for 24 h prior to fixation.

**Adipocyte fixation and nuclei collection**. We rinsed 10 M adherent HWA with serum-free media prior to fixation. We fixed the HWA directly in culture plates with 2% formaldehyde (EMD Millipore 344198) in serum-free adipocyte nutrition media (PromoCell). We incubated the cells in fixation medium with rocking at room temperature for 1 min, and then quenched with 1 M ice-cold glycine for a final concentration of 125 mM. After 5 min of rocking incubation at room temperature, we rinsed fixed cells twice with ice-cold PBS. Then we incubated the cells with rocking on ice with ice-cold permeabilization buffer (10 mM Tris–HCl pH 8, 10 mM NaCl, 0.2% Igepal CA-630, Complete EDTA-free protease inhibitor cocktail [Roche])[36] for 30 min. We scraped cells from the culture plate and centrifuged at $2500 \times g$ for 5 min at 4 °C to pellet nuclei. The supernatant was discarded and nuclei were flash frozen in liquid nitrogen and put at −80 °C.

**Hi–C library preparation**. We prepared the Hi–C library as described in Rao et al.[7] with modifications. We processed 10 M HWA nuclei in 5 M cell aliquots, closely following Rao et al.[7] protocol I.a.1 except we digested chromatin with 400U of *Hind*III (New England Biolabs R3104) at 37 °C overnight with shaking (950 rpm). After digestion, we pelleted nuclei with centrifugation at $2500 \times g$ for 5 min at 4 °C. We then resuspended nuclei in 265 μl 1× NEBuffer 2 and removed 10% of the cells and kept on ice for a 3 C control[37]. We followed Rao et al.[7] protocol I.a.1 to end-fill and mark with biotin, perform in-nucleus ligation, degrade protein, and perform ethanol precipitation and purification, except we used biotin-14-dCTP (Invitrogen 19518-018) to incorporate biotin during the end-filling step. After quality control to examine Hi-C marking and ligation efficiency, we sheared 5 μg of DNA to 250–550 bp using the Covaris M220 instrument. We performed double size-selection using Agencourt AMPure XP beads (Beckman Coulter A63881) as described in Rao et al.[7] protocol I.a.1.

We immobilized the fragments containing biotin using DYNAL™ MyOne™ Dynabeads™ Streptavidin T1 (Invitrogen 65601) beads following Rao et al.[7] protocol I.a.1. After end-repair and attachment of dATP, we ligated Illumina paired-end adaptors to the bead-bound library following the SureSelect^XT user manual for Illumina Paired-End Multiplexed Sequencing (Agilent Technologies). After washing, we resuspended the Hi–C library in 20 μl of 1× Tris buffer and

subsequently removed the Streptavidin beads from the DNA by heating at 98 °C for 10 min. We then amplified the adaptor-ligated library using 8 PCR cycles and purified using Agencourt AMPure XP beads, following the SureSelect^XT user manual.

**Capture Hi-C**. The RNA baits were designed in Mifsud et al.[6] for capturing *Hind*III fragments containing gene promoters (Dr. Cameron Osborne kindly shared the exact design). As described in Mifsud et al.[6], 120-mer RNA baits were designed to target both ends of *Hind*III fragments that contain annotated gene promoters (Ensembl promoters of protein-coding, noncoding, antisense, snRNA, miRNA and snoRNA transcripts). The bait sequence was deemed valid if GC content ranged from 25 to 65%, contained <3 consecutive Ns, and was within 330 bp of *Hind*III fragment ends. A total of 550 ng of the Hi–C library was hybridized to the biotinylated RNA baits, captured with DYNAL™ MyOne™ Dynabeads™ Streptavidin T1 beads, and amplified in a post-capture PCR to add indexes, using 12 PCR cycles. The library was sequenced on the Illumina HiSeq 4000 platform.

**Capture Hi-C data processing and interaction calling**. To ensure all downstream analysis was comparable, we first reduced the number of sequencing reads to match the number used in Mifsud et al.[6] analysis of their CHi-C data. We next processed the sequencing reads with the Hi–C User Pipeline (HiCUP) software[38], aligning reads to the human reference genome (GRCh37/hg19) and using all HiCUP default parameters. We called significant chromosomal interactions with the Capture Hi-C Analysis of Genome Organization (CHiCAGO) software[39], using default parameters, including the threshold of 5 for calling significant interactions. Briefly, the background is estimated by borrowing information across interactions on two separate components of the data: the interactions with baited fragments in the surrounding genomic region are used to model Brownian collisions, which are distance-dependent interactions, and interchromosomal interactions are used to model technical noise. CHiCAGO then employs a weighted *p*-value based on the expected number of interactions at a range of distances[39].

**Adipocyte nuclear protein extraction**. Nuclear protein was extracted from adipocytes after centrifugation of cells at $200 \times g$ for 5 min using a nuclear protein extraction kit as recommended (Active Motif 40010). The quantity of protein extracted was measured with BCA protein assay kit (Pierce 23227).

**Electrophoretic mobility shift assay**. Oligonucleotide probes (15 bp flanking SNP site for reference or alternate allele) (Supplementary Table 10) with a biotin tag at the 5′ end of the sequence (Integrated DNA Technologies) were incubated with HWA nuclear protein and the working reagent from the Gelshift Chemiluminescent EMSA kit (Active Motif 37341). For competitor assays, an unlabeled probe of the same sequence was added to the reaction mixture at 100 × excess. The reaction was incubated for 30 min at room temperature, and then loaded on a 6% retardation gel (ThermoFisher Scientific EC6365BOX) that was run in 0.5 × TBE buffer. The contents of the gel were transferred to a nylon membrane, and visualized with the chemi-luminescent reagent as recommended. Similarly, we performed the EMSAs with 1 μg purified CTCF protein (Origene TP720882). Supershift assays were performed with 1 μg anti-CTCF antibodies (Santa Cruz sc-15914 and EMD Millipore 07–729) that were incubated on ice with nuclear protein from HWA for 30 min prior to addition of oligonucleotide probes and run on gel.

**Study cohort**. The study sample consisted of a subset of the participants of the Finnish Metabolic Syndrome in Men (METSIM; $n = 10,197$) cohort, described in detail previously[40,41]. The study was approved by the local ethics committee (Research Ethics Committee, Hospital Restrict of Northern Savo) and all participants gave a written informed consent. The METSIM participants are Finnish males recruited at the University of Eastern Finland and Kuopio University Hospital, Kuopio, Finland. The median age of the METSIM participants is 57 years (range: 45–74 years). The biochemical lipid, glucose, and other clinical and metabolic phenotypes were measured, as described previously[40,41]. A random subset of the METSIM men underwent a subcutaneous abdominal adipose needle biopsy, with 335 unrelated individuals (IBD sharing estimated as <0.2 using a genetic relationship matrix calculated in PLINK[42]) analyzed here using RNA-seq.

**Identification of *cis*-eQTL SNPs**. We processed the METSIM RNA-seq dataset similarly as described in Rodriguez et al.[43]. Briefly, for the METSIM RNA-seq dataset, we isolated total RNA from abdominal subcutaneous adipose needle biopsy using the Qiagen miRNeasy kit. Polyadenylated mRNA was prepared using the Illumina TruSeq RNA Sample Preparation Kit v2 and sequenced using Illumina HiSeq 2000 platform generating paired-end, 50-bp reads. We used STAR[44] to align the reads to the hg19 reference genome, and assembled transcripts using Cufflinks v2.2.1[45]. We filtered genes for those with expression of FPKM > 0 in more than 90% of the samples. Additional details of this dataset have been previously described in Rodriguez et al.[43]. We inverse-normal transformed the FPKMs and adjusted for hidden confounding factors using PEER[46] by removing 22 PEER

factors based on a *cis*-eQTL analysis on chromosome 20 and choosing an optimal number of PEER factors without a loss of statistical power.

To decrease computation time, we prephased the METSIM genotype data, produced using the Illumina HumanOmniExpress BeadChip, by employing SHAPEIT2[47] with the phase 1 version 3 reference panel of the 1000 Genomes Project. We performed imputations with the same reference panel and IMPUTE2[48] with a cosmopolitan imputation approach, which included all populations from the 1000 Genomes Project, to ensure a high accuracy and maximize the number of imputed SNPs. Imputed data were filtered using the quality control inclusion criteria of info ≥0.8, MAF ≥5%, and Hardy–Weinberg equilibrium (HWE) $p >$ 0.00001. The *cis*-eQTL analysis was performed using Matrix-eQTL[49]. We classified the variants as in *cis* if they were within 1 Mb of either end of a gene. The first three genetic principal components were included as covariates in the *cis*-eQTL analysis to account for population stratification.

**Replication of *cis*-eQTL analysis in GTEx.** To ensure robustness of the results, we filtered the *cis*-eQTL SNPs and their target genes detected in the METSIM cohort so that both the *cis*-eQTL SNP and its predicted target gene were replicated in the *cis*-eQTL data by the GTEx Consortium ($n = 277$) for subcutaneous adipose tissue, filtered using their permutation test for significance, which used the adaptive permutation scheme in FastQTL[50] and a permutation test *p*-value threshold equal to the empirical *p*-value of the gene closest to the FDR 5% threshold, as reported by GTEx[12]. Only replicated adipose *cis*-eQTLs and their target genes were used in our downstream analyses.

**Heritability of *cis* expression in chromosomal interactions.** To investigate the functional importance of open chromatin regions (i.e., DHSs) within chromosomal interactions in adipocytes to heritability of *cis* expression, we used LD-score regression[11]. More specifically, we generated an annotation for each region within 1 Mb of the TSS of every gene with at least 1 significant promoter interaction. Per gene, this annotation consists of marking the variants within a distal fragment within 1 MB of the TSS that interact with the fragment containing the promoter of the gene. We further refined these annotations to the open chromatin regions available for TF binding. Accordingly, we only marked those variants located in regions identified in the union of DNase I hypersensitivity sites (DHSs) from all tissues in the ENCODE and Roadmap Epigenomics project[51]. Since these chromosomal interaction annotations change on a per-gene basis, we could not use the genome-wide overlapping matrix in the original software, which treats the annotations as fixed genome-wide. In our analyses, we generated an average overlapping matrix aggregated across all the regions. Importantly, we tested that this weighted overlap matrix does not qualitatively change the overall enrichment of heritability of gene expression for fixed annotations, such as coding regions (Supplementary Figure 1). These changes amount to altering equations 7 and 8 from Liu et al.[11] as follows (Equation 1 and 2).

Equation 1: Modified equation 8 from Liu et al.[11] using a weighted overlap matrix instead of the genome-wide average.

$$\mathrm{prop}_{h_g^2(C)} = \frac{h_g^2(C)}{h_g^2(\mathrm{total})} = \frac{\sum_C \overline{\tau_C}\,\overline{M}_{C' \cap C}}{\sum_C \overline{\tau_C}\,\overline{M}_C}$$

$$\mathrm{Where}\ \overline{M} = \sum_{\mathrm{gene}\ i}^{N} \frac{M_i}{NSNP_i}$$

In the equation above, C is a given annotation category; $\mathrm{prop}\_h_g^2(C)$ is the proportion of heritability for a given category; $\overline{\tau_c}$ is the regression coefficient for the category; $\overline{M}$ is the average overlap matrix for each local region; $M_i$ is the overlap matrix for each gene in the local region; and $NSNP_i$ is the number of SNPs in each local region.

Equation 2: Modified equation 9 from Liu et al.[11] using the average proportion of SNPs instead of the genome-wide average.

$$\mathrm{enrichment}(C) = \frac{\mathrm{prop}_{h_g}^2(C)}{\mathrm{prop}_{SNP_s}(\overline{M})}$$

$$\mathrm{Where}\ \overline{M} = \sum_{\mathrm{gene}\ i}^{N} \frac{M_i}{NSNP_i}$$

In the equation above, the variables are as in Equation 1, and $\mathrm{prop}\_SNPs(\overline{M})$ is the proportion of SNPs in the overlap matrix for a given category.

**Transcription factor motif enrichment in adipocytes.** We used Hypergeometric Optimization of Motif EnRichment (HOMER, v4.9) to investigate the enrichment of known TFs in the open chromatin regions (i.e., DHSs) within chromosomal interactions in adipocytes[9]. As input data, we used chromosomal interactions in adipocytes that interacted with a promoter fragment intersected with the union of all DHSs from ENCODE and Epigenomics Roadmap. We chose to use the DHSs in all cell types since there are no publicly available DHS data in adipocytes or adipose. Furthermore, since we were interested in the TF enrichments in adipocytes, we used CD34[+] chromosomal interactions intersected with the union of all DHSs as the background input file[6]. Any regions that were shared between the CD34[+] and adipocyte datasets were not considered in this analysis. We considered significant any TFs that were enriched in the DHSs within chromosomal interactions in adipocytes at an FDR of 5%. To ensure our background input file was not biasing the results, we also performed the same analysis with all DHSs not found in adipocyte chromosomal interactions as the background input.

We also assessed predicted differential TF binding using the tool deep learning–based sequence analyzer (DeepSEA)[19], which assesses differential histone modification, TF binding, and DHS profiles using a deep learning-based algorithmic approach and gives a functional significance score at the single nucleotide resolution.

**Overlap of *cis*-eQTL SNPs and chromosomal interactions.** To investigate functional *cis*-eQTL SNPs, we overlapped the imputed *cis*-eQTL SNPs and their target genes with Capture Hi-C chromosomal interactions by first overlapping the position other end of the looping interaction with the location of the *cis*-eQTL SNP. These were subsequently designated as regulatory element *cis*-eQTL SNPs. Simultaneously, we examined the identity of the predicted target gene for the *cis*-eQTL SNP and the gene involved in the looping interaction for a match. Only when both these criteria were fulfilled, was the *cis*-eQTL SNP defined as a looping *cis*-eQTL SNP and considered for further analyses.

**Identification of LD proxies of GWAS SNPs.** GWAS variants associated with BMI were obtained from Locke et al.[2], and with lipids and metabolites from Willer et al.[15] and Shin et al.[14]. We identified the *cis*-eQTL SNPs in tight LD ($r^2 > 0.80$) with GWAS variants in the METSIM adipose RNA-seq dataset using PLINK[42] and used them as the LD proxies for BMI, lipid, and metabolite GWAS SNPs. These sets of *cis*-eQTL SNPs were considered as BMI GWAS SNPs, lipid GWAS SNPs, and metabolite GWAS SNPs, respectively. These set of BMI, lipid, and metabolite GWAS SNPs were then overlapped with the looping *cis*-eQTL SNPs to identify all BMI, lipid, and metabolite GWAS SNPs involved in chromosomal interactions acting through distant regulatory elements.

**Correlation of BMI with adipose gene expression.** The BMI measurements in the METSIM cohort were first adjusted for age, age² and then the resulting residuals were inverse normal transformed to reduce the possible outlier effects. Next, we log transformed the FPKM values and then corrected them for 14 technical factors, including the RIN values, batch, percentage of coding reads, 5′ to 3′ bias, and percentage of uniquely mapped reads using Picard tools. The expression levels were correlated with the BMI measurements using Pearson correlation. The *p*-values were corrected for multiple testing for the number of eGenes using the Bonferroni correction (adjusted *p*-value < 0.05). To directly compare the effects sizes and *p*-values for BMI associations in TwinsUK with those in METSIM, we also tested the 42 replicated genes using a linear regression model with BMI and age, age² and the 14 technical factors as covariates when compared to a null model without BMI in METSIM (Table 1 and Supplementary Table 6). These models were compared using an F-test.

**Replication of BMI-adipose gene expression correlation.** Association analysis between BMI and adipose expression in the TwinsUK cohort was performed on 720 female twins. RNA-seq was generated as previously described[52] and gene level quantifications were generated to Gencode v19. Association between gene expression level and inversed normalized BMI was tested with a linear mixed effects model (LMEx) implemented using the lme4 package[53]. A full model including BMI was compared to a null model in which the same model was fitted, but with BMI omitted. These models were compared using an F-test. All known technical variables (batch, GC content, insert size mode, and primer index), age, age², and family structure were included as covariates in the models. All variables were centered and scaled to unit variance. Four genes were not present in the TwinsUK cohort dataset and we were thus unable to test them for replication, resulting in 54 genes tested for replication. Each replicated gene was examined to determine if effect size direction in TwinsUK and METSIM has the same direction. A Bonferroni corrected *p*-value (adjusted $p < 0.001$) with the same direction of effect as in METSIM was considered as statistical evidence for replication in the TwinsUK.

**Data availability.** The human primary adipocyte Capture Hi–C data are available at GEO (Accession ID: GSE110619).

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

## Acknowledgements

We thank the individuals who participated in the METSIM and GTEx studies. We also thank the sequencing core at UCLA for performing the RNA sequencing. In addition, we thank Cameron Osborne for his advice with the CHi-C protocol. We thank Xuanyao Liu for her assistance with the LD Score software. Francis Collins is thanked for providing the METSIM genotype data. The Genotype-Tissue Expression (GTEx) Project was supported by the Common Fund of the Office of the Director of the National Institutes of Health, and by NCI, NHGRI, NHLBI, NIDA, NIMH, and NINDS. The data used for the analyses described in this manuscript were obtained from: the GTEx Portal on 03/23/17. This study was funded by National Institutes of Health (NIH) grants HL-095056, HL-28481, U01 DK105561, R00 HL121172, and DK093757. D.Z.P. was supported by the NIH-NCI National Cancer Institute grant T32LM012424, M.A. was supported by the NIH grant T32HG002536, and A.K. by NIH grant F31HL127921. The funders had no role in study design, data collection and analysis, decision to publish, or preparation of the article. Genotyping for the METSIM cohort were supported by NIH grants DK072193, DK093757, DK062370, and Z01HG000024 and provided by the Center for Inherited Disease Research (CIDR). CIDR is fully funded through a federal contract from the NIH to The Johns Hopkins University, contract number HHSN268201200008I.

## Author contributions

D.Z.P., K.M.G., P.P., and A.K designed the study. D.Z.P., K.M.G., J.B., P.P., A.K., R.M.C., J.S.S., and Z.M. performed methods development and statistical analysis. D.Z.P., K.M.G., M.A., Z.M., and J.B. performed computation analysis of the data. K.M.G. and Y.V.B performed the experiments. A.K., E.N., M.A., K.L.M., C.R., and P.P. performed RNA-sequencing and quality control. M.L. performed phenotyping. M.C., A.J.L., M.L., E.N., K.

L.M., M.B., and P.P performed data collection and METSIM genotyping. C.A.G. and K.S.
S performed the replication analysis (TwinsUK). D.Z.P., K.M.G., A.K., and P.P. wrote the
manuscript and all authors read, reviewed, and/or edited the manuscript.

## Additional information

018-03554-9.

**Competing interests:** The authors declare no competing interests.

**Reprints and permission** information is available online at http://npg.nature.com/
reprintsandpermissions/

**Publisher's note:** Springer Nature remains neutral with regard to jurisdictional claims in
published maps and institutional affiliations.

