## [peer review file · Nature Communications]

Reviewer #1 (Remarks to the Author):

This is a thorough investigation into the enrichment of genetic variants associated with BMI and related traits in critical epigenetic signatures in adipocytes. The strengths of the paper include the use of primary adipose tissue cells to perform the HiC capture experiments to identify regions of the genome important to the function of adipocytes. The strengths are finding that the epigenetic marks are enriched for transcription factor binding sites for two critical adipose transcription factors, PPARG and CEBPB and SNPs contributing to heritability. The most well known adipose transcription factor, PPARG, only has a p value of 0.01 however, and it is not totally clear how likely it is that this could have occurred by chance, given the number of tests performed.

The main findings are that 42 genes have a set of interesting epigenetic and genetic features that make them very worthy of follow up. I would argue that these might be very interesting for adipose tissue biology, but not necessarily BMI, two very different things. One of the features the authors use, and replicate in the UKTwins RNAseq data, is that the gene expression of these 42 genes correlates with BMI in adipose tissue. However, previous papers (Emilsson et al al Decode) have shown that the adipose gene expression of more than 50% of all human genes correlates with BMI, so this is not much of an enrichment factor. More important is whether the eQTL SNPs are associated with a relevant trait, and here only 1 of the 4 variants the authors focus on, is associated with BMI. The other 3 are not associated with BMI in available GWAS data.

My main criticism is the ambiguous use of the phrases “obesity genes” or “BMI genes” and “important for BMI”. I don’t think the enrichment data, interesting and important though they are, prove that any of the genes are “obesity genes”. This term is in itself ambiguous and strictly speaking would require direct functional evidence from a mutation in the gene or manipulation of the gene in an animal model. It is important to note the difference between importance to BMI and importance to adipose tissue biology.

Other points:

1. The abstract contains some statements that need to be altered or qualified in my opinion – related to my points above. Including “obesogenic loci”.
2. Introduction. The SNP based heritability of BMI may be about 21% but that is not the same as all common genetic variants, it is just about the SNPs and variation captured on that array.
3. Because the results come before the methods, it is worth either simplifying some statements, or explaining more. For example it is not clear what the HindIII fragments are without going to the methods.
4. Is the enrichment of PPARG TFBS in DHSs noteworthy after correction for multiple testing ? It only reaches ≈ 0.01 and there were 26 /332 TFBS’s enriched? Similarly, the SNPs in adipocyte DHS explain 4.6% heritability compared to 0.23% expected, but does this account for the fact that DHS will be closer to genes ?

5. How does HOMER software work? Does it correct for the likely enrichment of DHS near genes and therefore TFBS ?
6. In the eGenes section of the results, the narrative is hard to follow in places. For example, I am not clear where the 386K eQTLs come from when first mentioned. Would it not be easier on the reader to clump these into independent signals and talk about the 100s-1000s of independent eQTLs into associated SNPs, which could include 100s in strong LD with the index (most strongly associated) variant?
7. In the results section on “looping” and elsewhere in the text, I think the term “BMI genes” is being used too loosely. This is a very ambiguous term that could refer to a gene whose expression level is correlated with BMI, a gene lying in a functionally interesting region relevant to BMI (such as DHS) or a SNP associated with BMI. Please be clearer – see main comments.
8. Whilst the MAP2K gene story is nice, I think it is too strong to say that this data confirms it as a “BMI gene”. That would require a coding mutation in a human, or manipulation in an animal model. Note that the adipose tissue gene expression of most genes in the genome correlates with BMI (Emilsson et al)

Reviewer #2 (Remarks to the Author):

Genome-wide association studies (GWASs) aim at deciphering the roles of variants such as single nucleotide polymorphisms (SNPs) in complex diseases. However the SNPs identified by GWASs can only explain a small part of the genetic heritability of these diseases. Moreover it is difficult to identify the causal SNPs among the SNPs that are associated to the diseases due to high linkage disequilibrium (LD) among close SNPs. By integrating GWAS SNPs with eQTLs and long-range chromatin interactions, Pan et al. identified SNPs that might act through long-range contacts with the four genes : MAP2K5, ORMDL3, LACTB and ACADS. These genes are associated with body mass index (BMI) and other obesity phenotypes. For instance, they show that the SNP rs4776984 is in long-range contact with the gene MAP2K5, it is an eQTL of MAP2K5, and it is in almost perfect LD with a BMI GWAS SNP (rs16951275). Moreover they demonstrate that the SNP rs4776984 can increase to predicted binding for CTCF protein, and using electrophoretic mobility shift assays of nuclear protein extracts, they observed increased protein binding for the SNP allele compared to the reference allele.

My major comments are the following:

- 1) The article is hard to read and to understand, especially for the subsections "Characterization of the adipocyte chromosomal interactions" and "Chromosomal interactions explain heritability of

gene expression". The article lacks clarity for multiple reasons. For instance, some terms are not explained such as "local gene expression", what does the term "local" refer to here? What is "partitioned LD Score Regression"? What is the aim of it? Another reason why the article lacks clarity is because the sentences are sometimes not well connected and some ideas are not introduced. Moreover, the authors must provide a schema to illustrate how data are integrated (Fig2/Sup Fig2 are really not enough). It is very difficult to understand clearly how data are integrated. Basically, the authors do multiple overlappings between sets such as eQTL SNPs, GWAS SNPs, LD SNPs, SNPs mapping to HindIII fragments, or eQTL genes, genes associated with BMI and genes mapping to HindIII fragments. The authors must plot multiple Venn diagrams (for instance) to represent the filtering procedures that were used to identify the candidate SNPs and genes that the authors identified.

2) In subsection "Chromosomal interactions explain heritability of gene expression", The authors show that DNase I hypersensitive sites (DHSs) in pChI-C non-promoter fragments are significantly enriched in the local gene expression. They must compare it to the DHSs not in non-promoter fragments in order to demonstrate the importance of long-range interacting loci in the regulation of gene expression. Similarly, they show that the variants in DHSs in pChI-C non-promoter fragments account for 4.6% of the heritability while account for 0.23% of the SNPs per local gene region. The authors must compare to the variants in the DHSs not in non-promoter fragments. The authors will obtain 4 different counts to compute an exact Fisher's test. Also, in subsection "Characterization of the adipocyte chromosomal interactions", the authors show that non-promoter fragments are enriched for histone marks. The authors must also demonstrate the enrichment for DHSs. Moreover the authors should compare the enrichments of DNA motifs in non-promoter fragment DHSs with other DHSs.

3) In subsection "Looping eGenes dissect novel GWAS genes for obesogenic traits", the authors found that the SNP rs4776984 showed an increased binding for CTCF, p300, RAD21 and SMC3. The authors must remove results for RAD21 and SMC2 because those proteins are part of the cohesin complex that cannot bind directly to the DNA. Instead cohesin is recruited by CTCF to the chromatin to form chromatin loops. Predicting the binding for CTCF is thus enough. Moreover, the authors can use two different tools: DeepSEA (<http://deepsea.princeton.edu/>) and gkmSVM (<http://www.beerlab.org/gkmsvm/>) to predict the impact of the SNP on DHS, CTCF binding and histone marks. In addition, the authors must do a ChIP of CTCF when doing the EMSA assay and show a supershift due to the antibody bound to CTCF.

4) The authors can validate the SNP rs4776984 if they demonstrate that it has a differential looping effect. For this purpose, the authors can do a 3C-qPCR experiment in one or more patient(s) carrying the SNP (rs4776984) and compared it to another/other patient(s) not carrying the reference allele. The 3C-qPCR should be designed to capture the long-range contact between rs4776984 and the promoter of MAP2K5.

5) The authors must provide more details about the pChI-C experiment results they obtained, such as basic statistics. How many reads were mapped? How many interactions were identified as significant? Is there any replicate to estimate the reproducibility?

Reviewer #3 (Remarks to the Author):

This is an interesting manuscript where complementary approaches for global interrogation of functional elements in the human genome is used to define (likely) functional SNPs controlling adipose gene expression. In addition, expression of these genes is correlated with BMI. I have the following comments

- It is not clear why GWAS of BMI, lipids, and metabolites in peripheral blood were used to define clinical implications of the findings. How important is adipose tissue, as compared to other organs, to determine metabolites in peripheral blood? The importance of primary disturbances in adipose tissue for controlling BMI is also unclear; most candidate genes for BMI from GWAS are primarily expressed in the brain. One reasonable hypothesis is that adipose gene expression is more important for body fat distribution and insulin resistance. Why were GWAS of these traits not investigated?
- The authors draw too far-reaching conclusions as regards the metabolite loci when discussing them in relation to obesity. These are metabolic traits, but the link to obesity in humans has not been established.
- The adipocyte pChIP-C DHS loci were overrepresented in adipocytes as compared to other cell types. But what about the corresponding eQTLs, are they specific to adipose tissue? If not, how to explain this?
- Table 2 is misleading as the number of genes per pathway is no more than 2.
- Are ORMDL3 or LACTB expressed in human adipocytes at the protein level? Do they have any function in human fat cells?
- The causal link between adipose tissue gene expression and BMI is unclear. I do not mean that the authors need to define this relationship. However, they should be more cautious in their writing about obesity based on presented data.

Details:

- In the Introduction it is written that “deep clinical phenotype data” were used; which data is referred to? Are there really “deep clinical phenotype data” in this study?
- “Chromosomal interactions explain heritability of gene expression” – This paragraph is difficult to understand. I do not understand what is meant by heritability is partitioned into 52 categories?

Responses to Reviewer 1:

We thank the Reviewer for the helpful critique and comments of the manuscript and have addressed all of the issues that were raised. We hope that the revisions satisfactorily respond to all of the Reviewer's concerns.

Reviewer #1 (Remarks to the Author):

This is a thorough investigation into the enrichment of genetic variants associated with BMI and related traits in critical epigenetic signatures in adipocytes. The strengths of the paper include the use of primary adipose tissue cells to perform the HiC capture experiments to identify regions of the genome important to the function of adipocytes. The strengths are finding that the epigenetic marks are enriched for transcription factor binding sites for two critical adipose transcription factors, PPARG and CEBPB and SNPs contributing to heritability. The most well known adipose transcription factor, PPARG, only has a p value of 0.01 however, and it is not totally clear how likely it is that this could have occurred by chance, given the number of tests performed.

My main criticism is the ambiguous use of the phrases “obesity genes” or “BMI genes” and “important for BMI”. I don't think the enrichment data, interesting and important though they are, prove that any of the genes are “obesity genes”. This term is in itself ambiguous and strictly speaking would require direct functional evidence from a mutation in the gene or manipulation of the gene in an animal model. It is important to note the difference between importance to BMI and importance to adipose tissue biology.

Response: We would like to thank the Reviewer for this comment and have thoroughly revised the entire manuscript to omit the ambiguous terms “obesity genes”, “BMI genes”, and “important for BMI”. We specifically now mention implications for adipose tissue biology instead of implications for BMI. The changes are underlined in the revised manuscript (pages 1, 4, 8, 11, and 14).

1. The abstract contains some statements that need to be altered or qualified in my opinion – related to my points above. Including “obesogenic loci”.

Response: To address the Reviewer's concern, we revised the abstract to omit the ambiguous terminology, including “obesogenic loci” (page 2).

2. Introduction. The SNP based heritability of BMI may be about 21% but that is not the same as all common genetic variants, it is just about the SNPs and variation captured on that array.

Response: We have revised the Introduction to state that the SNP-based heritability estimate of 21% that we reported was obtained using all HapMap phase 3 SNPs (~1.5M SNPs) in 16,275 unrelated individuals (page 3).

3. Because the results come before the methods, it is worth either simplifying some statements, or explaining more. For example it is not clear what the HindIII fragments are without going to the methods.

Response: As suggested by the Reviewer, we have simplified and/or explained the pChi-C experiments (including *HindIII* fragments), HOMER analysis, and LD Score Regression analysis more carefully in the revised Results (pages 5-6). In regard to the *HindIII* fragments created in the pChi-C protocol, we first crosslink proteins to DNA and digest DNA with the *HindIII* restriction endonuclease prior to re-ligating the *HindIII* fragments so that regions that were originally distant on the linear genome are then ligated together. When reads are mapped back to the genome, they are aligned to

the *HindIII* fragments so that counts per fragment are obtained, representing the number of times two regions were detected to be interacting.

4. Is the enrichment of PPARG TFBS in DHSs noteworthy after correction for multiple testing ? It only reaches ≈ 0.01 and there were 26 /332 TFBS's enriched? Similarly, the SNPs in adipocyte DHS explain 4.6% heritability compared to 0.23% expected, but does this account for the fact that DHSs will be closer to genes ?

Response: We would like to clarify regarding the multiple testing of the enrichment of the PPARG TFBS in DHSs that the Benjamini Hochberg adjusted p -value after correcting for multiple testing is 0.039, which remains significant. Regarding the heritability of local gene expression explained by DHSs in adipocyte promoter-interacting regions, the 0.23% refers to the percentage of genome-wide SNPs that are located within these promoter-interacting DHSs. This small percentage accounts for 4.6% of the heritability of local gene expression. This enrichment analysis does not take into account the fact that DHSs are closer to genes. However, the other functional annotations we analyzed with LD Score are also enriched near genes (e.g. 5' UTR, coding regions, TSS, etc.), and it is thus not likely that the closeness to the genes biases the estimate of how much these 0.23% of genome-wide SNPs explain of local gene expression.

5. How does HOMER software work? Does it correct for the likely enrichment of DHS near genes and therefore TFBS ?

Response: We thank the Reviewer for this comment and have revised the Results and Methods to clarify how the HOMER software works (pages 6, 29). Briefly, the HOMER software (Heinz et al. Mol Cell 2010) takes a set of input target and background regions and then calculates the number of times a TF motif is seen in the target and background sequences. Our target regions consisted of DHSs in promoter-interacting fragments in adipocytes that are not present in pChI-C data produced in CD34⁺ blood cells. The background is the vice versa: DHSs in CD34⁺ promoter-interacting fragments that are not present in our adipocyte data. The expected number of times a TFBS should be seen is calculated assuming the sites are evenly distributed across the genome. The proportion of times the TFBS is seen in the target when compared to the background is then tested for significant enrichment in the target sequences using the cumulative binomial distribution. We did not correct for the likely enrichment of DHSs near genes because both datasets are interrogating regions that are interacting with gene promoters and we assume that the promoter-interacting fragments come from a similar distribution of distances to genes.

6. In the eGenes section of the results, the narrative is hard to follow in places. For example, I am not clear where the 386K eQTLs come from when first mentioned. Would it not be easier on the reader to clump these into independent signals and talk about the 100s-1000s of independent eQTLs nto associated SNPs, which could include 100s in strong LD with the index (most strongly associated) variant?

Response: As suggested by the Reviewer, we have modified the Results to better describe our eQTL results (page 8). Specifically, we revised the Results and Supplementary Figure 2 to clearly show how we narrowed down to the 386k *cis*-eQTLs. Briefly, there are 487,679 SNPs, which were identified as *cis*-eQTLs in both METSIM and the GTEx adipose RNA-seq data. Among these 487,679 *cis*-eQTLs, the 386,068 *cis*-eQTLs have the same target gene and direction of effect in both METSIM and GTEx. We would also like to clarify that our design was to find the GWAS signals that regulate gene expression via chromosomal interactions as a looping *cis*-eQTL. Thus, we could not first do LD pruning of the *cis*-eQTL signals, and accordingly, we could not just focus on the strongest eQTL at each locus. We have further revised the Supplementary Figure 2 to clearly indicate the steps we used to overlap the eQTLs and pChI-C data to fine-map our *cis*-eQTLs.

7. In the results section on “looping” and elsewhere in the text, I think the term “BMI genes” is being used too loosely. This is a very ambiguous term that could refer to a gene whose expression level is correlated with BMI, a gene lying in a functionally interesting region relevant to BMI (such as DHS) or a SNP associated with BMI. Please be clearer – see main comments.

Response: We thank the Reviewer for this comment and have now omitted the ambiguous term “BMI genes” from the manuscript (pages 11 and 14).

8. Whilst the MAP2K gene story is nice, I think it is too strong to say that this data confirms it as a “BMi gene”. That would require a coding mutation in a human, or manipulation in an animal model. Note that the adipose tissue gene expression of most genes in the genome correlates with BMI (Emilsson et al)

Response: To address the Reviewer’s concern, we have now revised the wording to state that we have identified one possible candidate gene (page 12) and mechanism underlying the GWAS BMI signal at the *MAP2K5* locus. We have modified the section of Discussion on *MAP2K5* (page 17) to better describe the link between *MAP2K5* and adiposity. Briefly, *MAP2K5* is a member of the ERK5 MAP kinase signaling cascade, and the importance of ERK5 signaling in adipose was previously demonstrated in mouse *Erk5* knock-outs, which exhibit increased adiposity (Zhu et al. J Biol Chem 2014). This suggests that changes in ERK5 signaling in adipocytes could be relevant for human obesity. *MAP2K5* is a strong and specific activator of ERK5 in the ERK5 MAP kinase signaling cascade (Kato et al. EMBO 1997), supporting further study of *MAP2K5* in connection with increased adiposity.

Responses to Reviewer 2:

We thank the Reviewer for the helpful critique and comments of the manuscript and have addressed all of the issues that were raised. We hope that the revisions satisfactorily respond to all of the Reviewer's concerns.

Reviewer #2 (Remarks to the Author):

Genome-wide association studies (GWASs) aim at deciphering the roles of variants such as single nucleotide polymorphisms (SNPs) in complex diseases. However the SNPs identified by GWASs can only explain a small part of the genetic heritability of these diseases. Moreover it is difficult to identify the causal SNPs among the SNPs that are associated to the diseases due to high linkage disequilibrium (LD) among close SNPs. By integrating GWAS SNPs with eQTLs and long-range chromatin interactions, Pan et al. identified SNPs that might act through long-range contacts with the four genes : MAP2K5, ORMDL3, LACTB and ACADS. These genes are associated with body mass index (BMI) and other obesity phenotypes. For instance, they show that the SNP rs4776984 is in long-range contact with the gene MAP2K5, it is an eQTL of MAP2K5, and it is in almost perfect LD with a BMI GWAS SNP (rs16951275). Moreover they demonstrate that the SNP rs4776984 can increase to predicted binding for CTCF protein, and using electrophoretic mobility shift assays of nuclear protein extracts, they observed increased protein binding for the SNP allele compared to the reference allele.

My major comments are the following:

1. The article is hard to read and to understand, especially for the subsections "Characterization of the adipocyte chromosomal interactions" and "Chromosomal interactions explain heritability of gene expression". The article lacks clarity for multiple reasons. For instance, some terms are not explained such as "local gene expression", what does the term "local" refer to here? What is "partitioned LD Score Regression"? What is the aim of it? Another reason why the article lacks clarity is because the sentences are sometimes not well connected and some ideas are not introduced. Moreover, the authors must provide a schema to illustrate how data are integrated (Fig2/Sup Fig2 are really not enough). It is very difficult to understand clearly how data are integrated. Basically, the authors do multiple overlappings between sets such as eQTL SNPs, GWAS SNPs, LD SNPs, SNPs mapping to HindIII fragments, or eQTL genes, genes associated with BMI and genes mapping to HindIII fragments. The authors must plot multiple Venn diagrams (for instance) to represent the filtering procedures that were used to identify the candidate SNPs and genes that the authors identified.

Response: We thank the Reviewer for these comments and have revised the manuscript throughout to clarify all of the issues that the Reviewer lists here. Accordingly, we revised the 2 paragraphs (pages 7-8); explained the terms "local" and "partitioned LD Score Regression" (pages 6-7); and clarified the integration of the data analysis and how we overlapped the data sets (the revised Supplementary Figure 2 and page 8).

2. In subsection "Chromosomal interactions explain heritability of gene expression", The authors show that DNase I hypersensitive sites (DHSs) in pCHI-C non-promoter fragments are significantly enriched in the local gene expression. They must compare it to the DHSs not in non-promoter fragments in order to demonstrate the importance of long-range interacting loci in the regulation of gene expression. Similarly, they show that the variants in DHSs in pCHI-C non-promoter fragments account for 4.6% of the heritability while account for 0.23% of the SNPs per local gene region. The authors must compare to the variants in the DHSs not in non-promoter fragments. The authors will obtain 4 different counts to compute an exact Fisher's test. Also, in subsection "Characterization of the adipocyte chromosomal interactions", the authors show that non-promoter fragments are enriched for histone marks. The authors must

also demonstrate the enrichment for DHSs. Moreover the authors should compare the enrichments of DNA motifs in non-promoter fragment DHSs with other DHSs.

Response: As suggested by the Reviewer, we included a new functional category to our LD-score regression analysis to also compute the enrichment in local gene expression of SNPs located within DHSs while excluding the DHSs in the pChI-C non-promoter fragments (revised figure 1). Noteworthy, only a very small fraction of DHSs is excluded in this new DHSs category when compared to the overall DHSs category that we showed previously in Figure 1.

As suggested by the Reviewer, we then compared the enrichment of the DHSs in pChI-C non-promoter fragments in the local gene expression (enrichment=20.3; SD+/-5.2), with the enrichment of this new DHS category that excludes the DHSs in the pChI-C non-promoter fragments (enrichment=2.28; SD+/-0.2) (revised Figure 1).

As also suggested by the Reviewer, we compared the result that the 0.23% of the SNPs in DHSs in pChI-C non-promoter fragments account for 4.6% of the heritability of adipose tissue gene expression in *cis* with the result that the 13.25% SNPs in the new DHS category, which excludes the DHSs in the pChI-C non-promoter fragments, account for 30.21% of the heritability of adipose tissue gene expression in *cis*.

Taken together, these data show that the enrichment for heritability with the 0.23% SNPs in the DHSs in pChI-C non-promoter fragments is ~10x higher when compared to the 13.25% SNPs in this new DHS category that excludes the DHSs in the pChI-C non-promoter fragments, emphasizing their role in regulating local gene expression.

In our previous version of the manuscript, we demonstrated an enrichment of histone marks in the pChI-C non-promoter fragments. As suggested by the Reviewer, we now also performed an enrichment analysis for the DHSs in the pChI-C non-promoter fragments. We see a similar, significant enrichment for the DHSs in the pChI-C non-promoter fragments as for the histone marks, and we revised the manuscript accordingly (page 5 and Supplementary Table 2).

We have also included our results from HOMER comparing DHSs in adipocyte promoter-interacting fragments to DHSs in all other non-promoter, non-promoter-interacting regions (page 6). The enriched TFBSs are largely the same as in our previous comparison between the DHSs in the promoter-interacting regions in adipocytes and those in CD34⁺ cells. The major difference is in the p-values, which are much more significant in this new analysis that the Reviewer suggested, comparing the adipocyte promoter-interacting DHSs to all DHSs in non-promoter, non-interacting fragments. For example, the PPARG binding site is enriched in our adipocyte promoter-interacting DHSs when compared to the CD34⁺ promoter-interacting DHSs, with a p-value of 1×10^{-2} , whereas in the comparison of adipocyte promoter-interacting DHSs to all other non-promoter, non-interacting DHSs in the genome, the PPARG binding site is enriched with a p-value of 1×10^{-6} . We thank the Reviewer for this helpful clarification.

3. In subsection "Looping eGenes dissect novel GWAS genes for obesogenic traits", the authors found that the SNP rs4776984 showed an increased binding for CTCF, p300, RAD21 and SMC3. The authors must remove results for RAD21 and SMC2 because those proteins are part of the cohesin complex that cannot bind directly to the DNA. Instead cohesin is recruited by CTCF to the chromatin to form chromatin loops. Predicting the binding for CTCF is thus enough. Moreover, the authors can use two different tools: DeepSEA (<http://deepsea.princeton.edu/>) and gkmSVM (<http://www.beerlab.org/gkmsvm/>) to predict the impact of the SNP on DHS, CTCF binding and histone marks. In addition, the authors must do

a CHIP of CTCF when doing the EMSA assay and show a supershift due to the antibody bound to CTCF.

Response: We thank the Reviewer for this comment and have now removed the statement that RAD21 and SMC3 binding is predicted to be increased. We also thank the Reviewer for the suggestion to use the DeepSEA tool to predict the impact of SNP rs4776984 on DHS, CTCF binding, and histone marks. Using DeepSEA (deep learning–based sequence analyzer), we examined the allelic effect on protein binding of rs4776984 and the 15 other looping cis-eQTLs of MAP2K5. Of these 16 looping cis-eQTLs, two variants passed functional significance score <0.05 using DeepSEA. Of the two, our candidate functional eQTL SNP, rs4776984, resulted in the most significant functional score (2.36×10^{-3}) and was the only variant passing the functional significance score <0.01 among the 16 variants. Thus, the DeepSEA result further supports the differential TF binding at the variant site rs4776984 among all possible looping cis-eQTLs at the MAP2K5 locus. We thank the Reviewer for this helpful comment and have now included these DeepSEA results in the revised manuscript (page 11).

We also performed the supershift experiment using the CTCF antibody and adipocyte nuclear extract, and repeated this experiment three times. We did not observe a supershift in these EMSAs (new Supplementary Figure 5). We also performed the supershift experiment using another CTCF antibody (EMD Millipore 07-729), which resulted in the same negative finding (data not shown). However, a supershift experiment may remain negative even in the presence of true TF binding if a complex instead of a single TF alone is required for the TF binding. We also directly tested the CTCF protein for allele-specific binding at rs4776984 using EMSA in 3 replicate experiments, and did not find evidence of sole CTCF protein binding (new Supplementary Figure 6). These supershift and CTCF protein EMSA results are now included in the revised Results (page 12 and new Supplementary Figures 5-6) and Methods (page 25). Even though we did not find evidence of CTCF binding at rs4776984, nevertheless our replicated EMSA experiments with adipocyte extract show a clear allele-specific binding of a protein at the rs4776984 site with the looping DHS and histone marks H3K27me3 and H3K9me3. As the DeepSEA tool identified multiple TFs binding the rs4776984 site in an allele-specific way (Supplementary Table 8), it will likely require testing of a large set of TFs in additional functional studies to identify the actual TF that binds this site. These studies are beyond the scope of the current study.

4. The authors can validate the SNP rs4776984 if they demonstrate that it has a differential looping effect. For this purpose, the authors can do a 3C-qPCR experiment in one or more patient(s) carrying the SNP (rs4776984) and compared it to another/other patient(s) not carrying the reference allele. The 3C-qPCR should be designed to capture the long-range contact between rs4776984 and the promoter of MAP2K5.

Response: We agree with the Reviewer that this experiment would be valuable to show that differential looping occurs within the nucleus. However, it is not feasible to establish and perform the suggested experiments within the time frame of the revision process because these studies are technically challenging and time consuming. Furthermore, to produce the adipocyte promoter Capture Hi-C data, we used a commercial cell line, and we do not have access to human cell lines with the particular genotypes at the candidate SNP site rs4776984, and moreover establishing a protocol for patient iPSC generation and differentiation into adipocytes in our lab is very time-consuming and outside the scope of this study. Instead, for additional validation of the looping interactions at the 4 GWAS loci, we now provide new evidence from our independent pCHI-C experiment in HWA with two replicates. These new pCHI-C data replicate the chromosomal interactions at all 4 GWAS loci, and thus further confirm our original findings (page 13 and new Supplementary Table 9), demonstrating the robustness of our pCHI-C results.

5. The authors must provide more details about the pChi-C experiment results they obtained, such as basic statistics. How many reads were mapped? How many interactions were identified as significant? Is there any replicate to estimate the reproducibility?

Response: As suggested by the Reviewer, we have now included details about the pChi-C data analysis in the revised Results (page 5 and Supplementary Table 1). As mentioned in our response to point 4 above, since the submission of this manuscript, we have repeated the pChi-C experiments in the same cell line, HWA, and found that the same GWAS SNP interactions reported in the current study were also identified in this subsequent experiment. This new data thus provide additional important evidence, supporting the conclusion that the interactions we report in the current study are robust. We revised the Results (page 13) and made a new Supplementary Table 9 to include the new pChi-C data.

Response to Reviewer 3:

We thank the Reviewer for the helpful critique and comments of the manuscript and have addressed all of the issues that were raised. We hope that the revisions satisfactorily respond to all of the Reviewer's concerns.

Reviewer #3 (Remarks to the Author):

This is an interesting manuscript where complementary approaches for global interrogation of functional elements in the human genome is used to define (likely) functional SNPs controlling adipose gene expression. In addition, expression of these genes is correlated with BMI. I have the following comments

-It is not clear why GWAS of BMI, lipids, and metabolites in peripheral blood were used to define clinical implications of the findings. How important is adipose tissue, as compared to other organs, to determine metabolites in peripheral blood? The importance of primary disturbances in adipose tissue for controlling BMI is also unclear; most candidate genes for BMI from GWAS are primarily expressed in the brain. One reasonable hypothesis is that adipose gene expression is more important for body fat distribution and insulin resistance. Why where GWAS of these traits not investigated?

Response: We thank the Reviewer for the comment and have now revised the manuscript (page 9) to clearly state the known importance of adipose tissue in energy homeostasis of the human body. We agree with the Reviewer that other organs also play a role in altering the traits investigated in this study, and accordingly, we have now revised the manuscript to explain that the goal of the current study is to dissect the contribution of adipose/adipocyte functions to these traits (page 9). As suggested by the Reviewer, we also added to the manuscript a list of GWAS studies that we investigated, including a type 2 diabetes and a waist-hip-ratio (WHR) adjusted for BMI GWASs (Shungin et al. Nature 2015 and Fuchsberger et al. Nature 2016), which did not result in any overlaps with our adipocyte pChI-C and adipose *cis*-eQTL data.

-The authors draw too far-reaching conclusions as regards the metabolite loci when discussing them in relation to obesity. These are metabolic traits, but the link to obesity in humans has not been established.

Response: We thank the Reviewer for this comment. In addition to the changes noted in our response above, we have now included a rationale of our decision to use the metabolite GWAS data into the revised Results (page 9). We would also like to clarify that these metabolic traits may not have been shown to be causal for obesity yet, but their relationship to the obese state has been studied previously, as described in the Discussion. Regarding the *LACTB* locus, a robust correlation between the levels of the succinylcarnitine metabolite and BMI has been shown in two independent cohorts, KORA and TwinsUK, previously (Suhre et al. Nature 2011). Additional support for *LACTB* as a causal gene for obesity derives from functional studies using transgenic overexpression of *Lactb* in mice, resulting in an increase in the fat-mass-to-lean-mass ratio (Yang et al Nat. Genet. 2009 and Chen et al. Nature 2008). These previous findings are discussed on pages 18-19 of the revised manuscript.

-The adipocyte pChI-C DHS loci were overrepresented in adipocytes as compared to other cell types. But what about the corresponding eQTLs, are they specific to adipose tissue? If not, how to explain this?

Response: We thank the Reviewer for the comment and would like to clarify that we observed that the TF motif enrichment within the DHSs (union of all publicly available DHS data) included TFs that are important for adipocyte function when compared to the DHSs in promoter-interacting fragments

from a different cell type (CD34⁺). To address the Reviewer's comment about the tissue-specificity of the looping *cis*-eQTLs, we used the GTEx *cis*-eQTL data for the 44 tissues here for the Reviewer as follows: To investigate whether the *cis*-eQTLs identified by adipose looping interactions for the 576 eGenes are enriched in the subcutaneous adipose tissue, we selected the strongest looping *cis*-eQTL per gene and ranked the *p*-values of each looping *cis*-eQTL in the 44 GTEx tissues in ascending order (please see the Figure below). For these ranking comparisons, we used summary statistics from publicly available GTEx *cis*-eQTL data. If a *cis*-eQTL was not significant in a given tissue, its rank was set at the shared highest rank. The figure below shows that overall the *cis*-eQTLs in the subcutaneous adipose tissue rank the lowest when compared to the other 43 tissues in the GTEx cohort. Using a t-test, the rank of looping *cis*-eQTLs in the subcutaneous adipose tissue was significantly lower than the rank of *cis*-eQTLs in the tibial nerve tissue, which was the second smallest ranking tissue (p -value = 1.27×10^{-9}) (please see the figure below). Similarly, the rank of the looping *cis*-eQTLs in the subcutaneous adipose tissue was also significantly lower than the ranks of the *cis*-eQTLs in the remaining 43 GTEx tissues (p -value < 2.2×10^{-16}). Taken together, these data suggest a potential enrichment of these looping *cis*-eQTLs in the subcutaneous adipose tissue when compared to the 43 other GTEx tissues. However, we recognize that we selected looping *cis*-eQTLs that are also significant *cis*-eQTLs in GTEx adipose tissue, which might inflate the enrichment signal. Therefore, another independent cohort is required to estimate the adipose-specificity of looping *cis*-eQTLs without bias. Since currently there is no other multitissue cohort available in addition to GTEx, future studies in independent multitissue cohorts are warranted to fully verify the adipose enrichment of these looping *cis*-eQTLs.

The box plots show the medians of the ranks of looping *cis*-eQTLs from our 576 eGenes in our study across all GTEx tissues. A significantly smaller rank of the *cis*-eQTLs was observed for the subcutaneous adipose tissue when compared to the tibial nerve tissue (p -value = 1.27×10^{-9}), and overall smaller when compared to the ranks of the *cis*-eQTLs in the remaining 42 GTEx tissues (p -value < 2.2×10^{-16}).

-Table 2 is misleading as the number of genes per pathway is no more than 2.

Response: We thank the Reviewer for this comment and have added information on the genes within each pathway to demonstrate that the genes do not fully overlap. To not overstate the importance of this finding that is based on small numbers, we moved it to the Supplementary results (Supplementary Table 7).

-Are ORMDL3 or LACTB expressed in human adipocytes at the protein level? Do they have any function in human fat cells?

Response: To address the Reviewer's comment, we have further examined protein-level expression and function of these two proteins in adipose tissue and/or adipocytes. While ORMDL3 protein expression is not available in public databases (e.g. The Human Protein Atlas), Ormdl3 is expressed in mouse adipocytes differentiated from 3T3-L1 cells. Furthermore, its expression at the protein level is used in this previous study as a readout of ER stress in adipocytes (Kajimoto et al. 2014). As the function of the ORMDL3 protein has not been studied in human primary adipocytes, we are thus limited to various BMI-related transcriptomic studies, which consistently show a significant inverse correlation of *ORMDL3* adipose expression with BMI (Heinonen et al. *Diabetologia* 2017; He et al. *Sci Rep* 2017), as reported in our study. We face a similar issue for *LACTB*, which has a limited number of studies on the endogenous protein function in adipocytes. A transgenic *Lactb* mouse model demonstrated an increased fat mass when compared to the WT mice; however the adipose tissue transcriptome was not studied. Nevertheless, as mentioned in our responses previously, functional studies using transgenic overexpression of *Lactb* in mice supports *LACTB* as a causal gene for obesity, because transgenic overexpression of *Lactb* in mice results in an increase in the fat-mass-to-lean-mass ratio. We have revised our Discussion (page 18) to note this lack of evidence at the protein level for both of these proteins and call for further molecular studies to determine the function of *ORMDL3* and *LACTB* in connection with obesity.

-The causal link between adipose tissue gene expression and BMI is unclear. I do not mean that the authors need to define this relationship. However, they should be more cautious in their writing about obesity based on presented data.

Response: We thank the Reviewer for this comment and have carefully revised the manuscript to avoid stating that our results provide a causal link to BMI. Instead we now state that our study furthers the understanding of adipose tissue and adipocyte biology and their role in obesity-related human phenotypes (page 4, 8, 11, and 14).

Details:

-In the Introduction it is written that “deep clinical phenotype data” were used; which data is referred to? Are there really “deep clinical phenotype data” in this study?

Response: To address the Reviewer's concern, we omitted this expression from the Introduction and instead clarified in the revised the Introduction and Results what exact metabolic clinical phenotypes we used in the METSIM cohort (pages 3 and 14).

-“Chromosomal interactions explain heritability of gene expression” – This paragraph is difficult to understand. I do not understand what is meant by heritability is partitioned into 52 categories?

Response: To address the Reviewer's comment, we rewrote this section to clearly describe how the LD Score Regression method was utilized in our study (page 7). Partitioning heritability to 52

categories is now explained as follows: To investigate whether the variants residing within the open chromatin in the adipocyte chromosomal looping regions are enriched for the heritability of *cis* expression regulation, we partitioned the heritability of *cis* regulation of human adipose gene expression into 52 functional categories using a modified partitioned LD Score regression method (Liu et al. Am J Hum Genet 2017 & see Methods). The 52 functional categories are derived from 26 main annotations that included coding regions, untranslated regions (UTRs), promoters, intronic regions, histone marks, DNase I hypersensitivity sites (DHSs), predicted enhancers, conserved regions, and other annotations (Liu et al. Am J Hum Genet 2017 & Supplementary Figure 1, Supplementary Tables 4-5). The partitioned LD Score regression method (Liu et al. Am J Hum Genet 2017) utilizes summary association statistics of all variants on gene expression to estimate how much variants in different annotation categories explain the heritability of *cis* and *trans* expression regulation while accounting for the LD among functional annotations. To assess the heritability enrichment by the variants in the chromosomal interactions detected by pChI-C on a per-gene basis, we further modified the LD score method, as described in detail in the Methods. We have now revised the Results to clarify this part (page 7).

Reviewer #1 (Remarks to the Author):

This remains a good paper tackling the very hard job of finding causal alleles and genes in a relevant tissue. In general I still think the authors are overselling their results, especially in the title. Whilst very interesting, I don't think the coincidence of one of the SNPs in an LD block representing a cis eQTL with a HiC looping feature means the functional SNP has been found. Other SNPs could be causal for different reasons and it is still not made clear whether or not these SNPs are the strongest of those associated with gene expression AND the strongest of those associated with the GWAS trait.

I still find it hard to follow some sections. This is perhaps reflected in the abstract, where it is hard to get a clear and simple message about the paper.

For example, the results section "identification of looping cis eQTL SNPs" is hard to follow. The section appears to be mis-titled in that it appears to be about the identification of genes not SNPs. The sequence appears to be

1. Take adipose cis eQTLs from metsim.
2. Take intersect of 1 above and the hiC promoter interacting regions.
3. Take 386k cis eQTLs consistent in direction in metsim and gtex. Is this a subset of 1 or 2 above? I think 1.? And these represent how many independent cis eQTLs given many will be in v strong LD?
4. these 386eQTLs are linked to 4332 genes. (Presumably the redundancy is mainly due to many tens of variants in LD per locus plus multiple signals per gene?)
5. Take the 576 / 4332 genes that occur in hiC identified regions (but how does this compare to step 2 above?)
6. Take the 52 / 576 genes where their adipose tissue expression correlates with BMI. This figure of ten % correlation with BMI seems very low given the adipose expression of most genes correlates with BMI.
7. Take the 42/ 52 genes where adipose tissue expression also correlates with BMI in another cohort - twins UK.

I think step 2 is confusing and the whole section could be clearer. It might be easier to simply list in one sentence the criteria for gene selection before explaining more. For example steps 1 and 3 could be "the genes selected had to a) have eQTLs in two studies, metsim with consistent directions in gtex", and steps 6 and 7 "b) have adipose expression correlated with BMI in the same direction in two studies".

I still have a problem with the next section “dissect novel GWAS genes for BMI “. The term “for BMI” is vague and gives the impression we know that they alter BMI when altered in some way. I don't think the authors have proven this. There is one cis eQTL in this paper that would justify being called a BMI SNP and the authors have done a good job at highlighting MAP2K5 as a strong candidate at this locus. (Note that it would be worth stating which gene the GIANT consortium had labelled this locus as, presumably not MAP2K5 as the index SNP is in a nearby gene?). Given the GWAS sample sizes, especially for BMI and WHR, a variant with an r^2 of 0.8 with the index GWAS variant could be very unlikely to be the causal variant compared to tens if not hundreds at stronger r^2 values. What are the r^2 values with the GWAS index SNP? The authors give this for MAP2K5 but not the other three.

To make the results clearer I suggest separate subheadings and paragraphs for each of the four GWAS genes discussed.

The final short paragraph of the results is not needed I feel. Once a gene is correlated with BMI it is not surprising it is going to be correlated with BMI related traits and doesn't help us decide if the cis gene-BMI link is real or not, and the results section is long and quite hard to follow, when it should be the clearest and quickest part of the paper to understand.

Other points:

1. It is still tough to get a clear and simple idea of the paper from the abstract. For example the second sentence is long and difficult to parse.
2. At the top of page six in the results, how do the authors define “more likely to be functional”?
3. You don't actually need to use the word “significant” anywhere. By stating a result you are implying it has reached a level of statistical confidence that makes it interesting, and significant gives the false impression it is definitely real.
4. Some discussion points creep into the results. For example the phrase “making the likely target gene.... difficult to establish”.
5. In general fig 5 and table 1 don't help the reader much I feel. As mentioned, the adipose expression of most genes is correlated with BMI, and therefore many BMI traits.

Reviewer #2 (Remarks to the Author):

The authors addressed my comments and the article has been significantly improved.

Reviewer #3 (Remarks to the Author):

The authors have give satisfactory answers to all my comments. I have no further questions.

Responses to Reviewer 1:

We thank the Reviewer for the helpful critique and comments of the manuscript and have addressed all of the issues that were raised. We hope that the revisions satisfactorily respond to all of the Reviewer's concerns.

This remains a good paper tackling the very hard job of finding causal alleles and genes in a relevant tissue. In general I still think the authors are overselling their results, especially in the title. Whilst very interesting, I don't think the coincidence of one of the SNPs in an LD block representing a cis eQTL with a HiC looping feature means the functional SNP has been found. Other SNPs could be causal for different reasons and it is still not made clear whether or not these SNPs are the strongest of those associated with gene expression AND the strongest of those associated with the GWAS trait.

Response: We thank the Reviewer for these comments. Regarding the function of the variants we report in the manuscript, we are not claiming that there are no other *cis*-eQTL SNPs in these regions affecting expression of the same gene. We would also like to clarify that the 4 reported looping *cis*-eQTL variants at the 4 GWAS loci represent either the actual GWAS variant, as is the case at the *ORMDL3* and *LACTB* GWAS loci, or are in perfect or very tight LD ($r^2=1.0$ or $r^2 \geq 0.98$, respectively) with the GWAS variants, as is the case at the *ACADS* and *MAP2K5* GWAS loci. However, the Reviewer is correct in that these and other GWAS variants do not necessarily represent the most significant *cis*-eQTL SNPs for any given gene. There are often multiple independent *cis*-eQTL variants driving expression with different effect sizes, and we are only identifying eQTL variants in looping regulatory elements for which we can postulate their mechanism of gene regulation. As we have pointed out in the revised Discussion (page 20), there may be other *cis*-eQTL SNPs that affect gene expression via other mechanisms than looping. Regarding the title, we would prefer keeping the current title, because we have now carefully addressed the potential overselling of our results throughout the text (please also see our responses below concerning this issue).

I still find it hard to follow some sections. This is perhaps reflected in the abstract, where it is hard to get a clear and simple message about the paper.

Response: We thank the Reviewer for the comment and have revised the abstract to make it clearer (page 2).

For example, the results section "identification of looping cis eQTL SNPs" is hard to follow. The section appears to be misstitled in that it appears to be about the identification of genes not SNPs. The sequence appears to be

- 1. Take adipose cis eQTLs from metsim.**
- 2. Take intersect of 1 above and the hiC promoter interacting regions.**
- 3. Take 386k cis eQTLs consistent in direction in metsim and gtex. Is this a subset of 1 or 2 above? I think 1.? And these represent how many independent cis eQTLs given many will be in v strong LD?**
- 4. these 386eQTLs are linked to 4332 genes. (Presumably the redundancy is mainly due**

to many tens of variants in LD per locus plus multiple signals per gene?)

5. Take the 576 / 4332 genes that occur in hiC identified regions (but how does this compare to step 2 above ?)

6. Take the 52 / 576 genes where their adipose tissue expression correlates with BMI . This figure of ten % correlation with BMI seems very low given the adipose expression of most genes correlates with BMI.

7. Take the 42/ 52 genes where adipose tissue expression also correlates with BMI in another cohort - twins UK.

I think step 2 is confusing and the whole section could be clearer . It might be easier to simply list in one sentence the criteria for gene selection before explaining more. For example steps 1 and 3 could be “the genes selected had to a) have eQTLs in two studies, metsim with consistent directoins in gtex”, and steps 6 and 7 “b) have adipose expression correlated with BMI in the same direction in two studies”.

Response: We thank the Reviewer for this useful comment and have now clarified the title of this section in the revised Results from “identification of looping cis eQTL SNPs” to “Identification of genes regulated by looping cis-eQTL SNPs”. We have also clarified in the revised Results that the intersect of *cis*-eQTLs from METSIM and the pChI-C interactions was taken after we replicated the *cis*-eQTLs in GTEx (page 8). This revision should clarify the previously confusing step 2 above.

I still have a problem with the next section “dissect novel GWAS genes for BMI “. The term “for BMI” is vague and gives the impressoin we know that they alter BMI when altered in some way. I don't think the authors have proven this. There is one *cis* eQTL in this paper that would justify being called a BMI SNP and the authors have done a good job at highlighting MAP2k5 as a strong candidate at this locus. (Note that It would be worth stating which gene the GIANT consortium had labelled this locus as , presumably not MAP2K5 as the index SNP is in a nearby gene?). Given the GWAS sample sizes , especially for BMI and WHR, a variant with an r^2 of 0.8 with the index gwas variant could be very unlikely to be the causal variant compared to tens if not hundreds at stronger r^2 values. What are the r^2 values with the gwas index snp ? The authors give this for MAP2k5 but not the other three.

Response: We thank the Reviewer for this comment, and have changed the subheading title from “Looping eGenes dissect novel GWAS genes for BMI and related metabolic traits” to “Adipocyte chromosomal interactions identify novel GWAS genes” (page 9).

Regarding the use of $r^2 > 0.8$ to define GWAS SNP proxies, we would like to clarify that the 4 looping variants we ultimately report are in very tight or perfect LD with the GWAS variant (or represent the index GWAS SNP itself) as follows:

- *ORMDL3* (the reported looping *cis*-eQTL variant is the index GWAS SNP) (page 12),
- *MAP2K5* (the reported looping *cis*-eQTL variant is in $r^2=0.98$ with the GWAS SNP) (page 11),

- *LACTB* (the reported looping *cis*-eQTL variant is the index GWAS SNP) (page 13), and
- *ACADS* (the reported looping *cis*-eQTL variant is in $r^2=1.0$ with the GWAS SNP) (page 13).

All r^2 values are reported in the Results. We have removed instances in which we discuss using $r^2>0.8$ when it is not necessary for understanding our methodology to clarify this point in the revised Results (pages 5 and 10). As mentioned above, we also stated in the Discussion that there may be other *cis*-eQTL SNPs that affect gene expression in the regions via other mechanisms than looping, or that may have more significant p -values (page 20).

To make the results clearer I suggest separate subheadings and paragraphs for each of the four GWAS genes discussed.

Response: As suggested by the Reviewer, we have now separated the GWAS results into 2 additional subsections to improve the clarity of the Results section (pages 10 and 12).

The final short paragraph of the results is not needed I feel. Once a gene is correlated with BMI it is not surprising it is going to be correlated with bmi related traits and doesn't help us decide if the cis e gene- bmi link is real or not, and the results section is long and quite hard to follow, when it should be the clearest and quickest part of the paper to understand.

Response: We thank the Reviewer for this comment and have removed both the final paragraph of the Results section and Figure 5 from the revised Results (page 14).

Other points:

1. It is still tough to get a clear and simple idea of the paper from the abstract. For example the second sentence is long and difficult to parse.

Response: We thank the Reviewer for the comment and have revised the abstract to make it clearer (page 2).

2. At the top of page six in the results, how do the authors define “more likely to be functional”?

Response: We thank the Reviewer for the comment and have changed this sentence to state that the open chromatin regions are more likely to bind transcription factors and thus be relevant for chromosomal looping interactions (page 6).

3. You don't actually need to use the word “significant” anywhere. By stating a result you are implying it has reached a level of statistical confidence that makes it interesting, and significant gives the false impression it is definitely real.

Response: We agree with the Reviewer and have removed various instances of the word “significant,” particularly when we already report the p -values (pages 6, 7, 15, and 38)

4. Some discussion points creep into the results. For example the phrase “making the likely target gene.... difficult to establish”.

Response: We thank the Reviewer for pointing this out and have removed this phrase from the revised Results (page 10).

5. In general fig 5 and table 1 don't help the reader much I feel. As mentioned, the adipose expression of most genes is correlated with BMI, and therefore many BMI traits.

Response: We thank the Reviewer for the comment and have removed both the final paragraph of the Results section and Figure 5 (page 14). We prefer to keep Table 1 in the manuscript to help the reader understand the approach we took to replicate our findings as well as highlight where the GWAS genes fall on this list.

Hello,

This is confirm that we have proof read the manuscript and made corrections.
We have also attached the responses to the query.

Thank you for your help.

Best,
Arthur

On 3/5/18 7:25 AM, jprod1.SN@adi-mps.com wrote:

Learn. Discover. Achieve

SPRINGER NATURE

Article Title : Integration of human adipocyte chromosomal interactions with adipose gene expression prioritizes obesity-related genes from GWAS

DOI : 10.1038/s41467-018-03554-9

NCOMMS-17-23684-T

Dear Author,

We are pleased to inform you that your paper is nearing publication. Your article proofs are available at:

http://eproofing.springer.com/journals_v2/index.php?token=lpvr8WH6UNJ_BwQUu0dwrGmj7fMaVYea

The URL is valid only until your paper is published online. It is for proof purposes only and may not be used by third parties.

You can help us facilitate quick and accurate publication by using our e.Proofing system. The system will show you an HTML version of the article that you can correct online. In addition, you can view/download a PDF version for your reference.

As you are reviewing the proofs, please keep in mind the following:

- This is the only set of proofs you will see prior to publication.
- Only errors introduced during production process or that directly compromise the scientific

integrity of the paper may be corrected.

- Any changes that contradict journal style will not be made.
- Any changes to scientific content (including figures) will require editorial review and approval.

Please check the author/editor names very carefully to ensure correct spelling, correct sequence of given and family names and that the given and family names have been correctly designated (NB the family name is highlighted in blue).

Please submit your corrections within 2 working days and make sure you fill out your response to any AUTHOR QUERIES raised during typesetting. Without your response to these queries, we will not be able to continue with the processing of your article for Online Publication.

Should you encounter difficulties with the proofs, please contact me.

Thank you very much.

Sincerely yours,

Springer Nature Corrections Team

MPS Limited,
HMG Ambassador, 137 Residency Road,
Bangalore - 560025, INDIA
e-mail: jprod1.SN@adi-mps.com
Fax: +91 80 4178 4222

SPRINGER NATURE
Learn. Discover. Achieve

 Springer
nature research

 BioMed Central
BioMed Central

 palgrave
macmillan
Apress

 SCIENTIFIC
AMERICAN
macmillan
education

 Springer Healthcare

 Springer Plivo
Springer Medline

- 1. We are pleased to inform you that this article will be indexed by the Nature Index (natureindex.com) approximately two months after final publication. Upon indexing, you will receive an automated email of notification from Nature Index. If you would prefer not to receive notification, please opt out by clicking the link:**

<http://www.natureindex.com/notifications/unsubscribe?email=a5ko@ucla.edu>

Thank you for the notification about Nature Index.

- 2. Author surnames have been highlighted - please check these carefully and indicate if the first name or surname have been marked up incorrectly. Please note that this will affect indexing of your article, such as in PubMed.**

We have confirmed that these are correct.

- 3. To ensure a streamlined publication process, Nature Communications offers only one round of proofs, and so this is the only opportunity you will have to make corrections to your paper prior to publication. You should ensure that you check the e-Proof carefully, coordinate with any co-authors and mark up all your final edits clearly before submitting your changes. Please note that, once you submit your corrections, we will be unable to make further changes to the Article.**

We acknowledge that there will only be a single round of final edits and proofs.

- 4. If you have any preprints in the reference list, please check whether these papers have been published and provide updated citation details.**

We do not have any preprints in the reference list.

- 5. To ensure clarity and avoid mistakes in the mathematics in your paper, please check all mathematical equations and symbols carefully for accuracy and consistency. Please also carefully check that units on quantities have been typeset correctly.**

We have checked all mathematical equations and symbols for accuracy and consistency.

- 6. Please check Figures for accuracy as they have been relabelled. (Please note that in the e-Proof the figure resolution will appear at lower resolution than in the pdf and html versions of your paper.)**

We have checked all of the Figures for errors.

- 7. If you wish to submit a revised Supplementary Information file, please provide the complete file and a full description of each of the changes you have made to this file, so that we can assess that they conform to journal guidelines. All replacement files should be uploaded into the e-Proof, or onto an FTP site, with the correct file name as listed in the e-Proof - e.g. 41467_2017_XXXX_MOESM1_ESM.pdf**

We have modified Supplementary Figure 7 to correct an error in the label. The correct label is "Purified CTCF protein". We uploaded the new Supplementary Information file and named it according to the guidelines above.